# WIND: Weather Inverse Diffusion for Zero-Shot Atmospheric Modeling

**Andreas Fürst** [*,2]    **Michael Aich** [*,1,4]    **Florian Sestak** [*,2,3]    **Carlos Ruiz-Gonzalez** [2]
**Niklas Boers** [1,4]    **Johannes Brandstetter** [2,3]

[1] Munich Climate Center and Earth System Modelling Group, TUM School of Engineering
and Design, Technical University of Munich, Germany
[2] ELLIS Unit, LIT AI Lab, Institute for Machine Learning, JKU Linz, Austria
[3] Emmi AI GmbH, Linz, Austria
[4] Potsdam Institute for Climate Impact Research, Potsdam, Germany
michael.aich@tum.de, fuerst@ml.jku.at

## Abstract

Deep learning has revolutionized weather forecasting, but many challenges remain, including climate modelling. Moreover, the current landscape remains fragmented: highly specialized models are typically trained individually for distinct tasks. To unify this landscape, we introduce WIND, a single pre-trained foundation model capable of replacing specialized baselines across a vast array of tasks. Crucially, in contrast to previous atmospheric foundation models, we achieve this without any task-specific fine-tuning. To learn a robust, task-agnostic prior of the atmosphere, we pre-train WIND with a self-supervised video reconstruction objective, utilizing an unconditional video diffusion model to iteratively reconstruct atmospheric dynamics from a noisy state. At inference, we frame diverse domain-specific problems strictly as inverse problems and solve them via posterior sampling. This unified approach allows us to tackle highly relevant weather and climate problems, including probabilistic forecasting, spatial and temporal downscaling, reconstruction of spatial fields from from sparse observations, and enforcing global dry air mass conservation purely with our pre-trained model. We further demonstrate how WIND can be applied to explore extreme weather events under prescribed out-of-distribution warming scenarios. By combining generative video modeling with inverse problem solving, WIND offers a computationally efficient paradigm shift in AI-based atmospheric modeling. Code is available at https://github.com/ml-jku/wind.

## 1 Introduction

Understanding atmospheric dynamics under climate change is of utmost importance. Adverse atmospheric conditions drive severe humanitarian and financial crises, with global economic costs exceeding $4.3 trillion over the past 50 years Swiss Re Institute (2024); World Meteorological Organization (2023). In terms of societal impact, precipitation is among the most critical variables. Extreme precipitation events lead to devastating floods and landslides, whose frequency and intensity increase with global warming IPCC (2023). Beyond disaster risk mitigation, atmospheric modeling is central to the energy transition. Wind speed, for instance, is a key predictor of power output and economic viability for renewable energy projects Yan et al. (2025).

Petabyte-scale atmospheric datasets, like ERA5 Hersbach et al. (2020), which offer decades of high-resolution data, are both a challenge and an opportunity for weather forecasting. While the predictive skill of classical numerical weather prediction (NWP) models scales primarily with increased computational power rather than historical dataset size Bauer et al. (2015), this data abundance directly fuels atmospheric foundation models Bodnar et al. (2025); Nguyen et al. (2023); Lessig et al. (2023). Pre-trained on massive datasets, these models capture the underlying dynamics and relationships of the atmosphere, encoding them into compact representations.

---

[*] Equal contribution

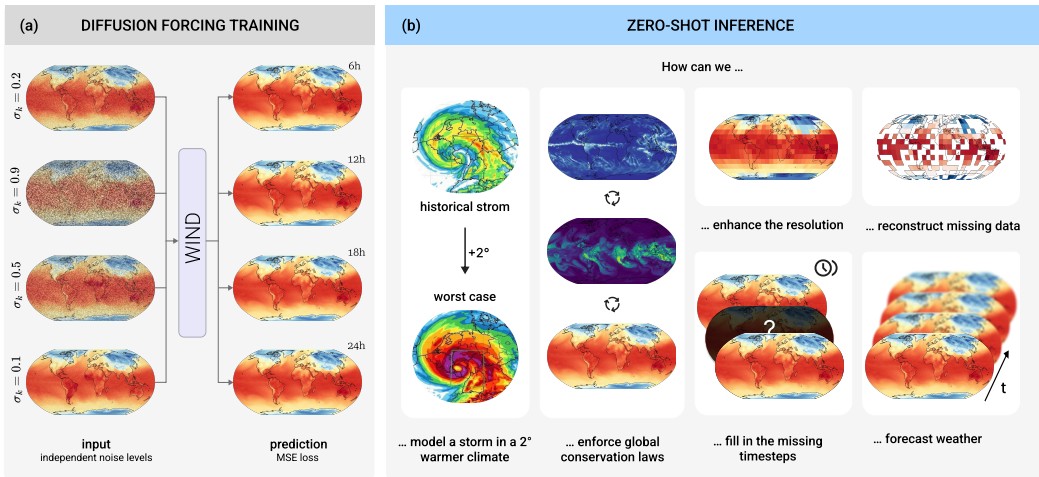

Figure 1: **Training setup and inference capabilities of WIND (a) Training:** We apply an independent noise level $\sigma_k$ to each frame in the sequence. The model is trained without explicit noise level information. WIND learns to jointly denoise the sequence, enabling it to handle arbitrary combinations of clean and noisy context frames. **(b) Inference:** WIND addresses various climate- and weather-related questions by framing them as inverse problems: recover the atmospheric field $x$ from the observation $y$ that is constraint by some operator $\mathcal{A}$ as $\mathcal{A}(x) = y$. We demonstrate how to formulate a task-specific operator $\mathcal{A}$ for each question in (b).

Foundation models mark a paradigm shift from the current fragmented landscape, where specialized models are trained from scratch for every niche (e.g., precipitation downscaling vs. wind forecasting). Instead, a single foundation model can be pre-trained once and efficiently fine-tuned to diverse downstream applications.

An effective atmospheric foundation model must reflect the chaotic and probabilistic nature of weather data. To address this, we present WIND, a framework that combines diffusion forcing training with moment matching posterior sampling (MMPS) at inference to construct a unified foundation model of the atmosphere. Unlike existing approaches, which train or fine-tune additional networks for each new task Bodnar et al. (2025); Nguyen et al. (2023); Lessig et al. (2023), we solve diverse downstream tasks purely at inference time. We use ERA5 data at 1.5° resolution, instead of 0.25°, to mitigate the prohibitive computational training costs. Thus, we focus on demonstrating the conceptual novelty and versatility of our approach, rather than competing directly with state-of-the-art operational baselines.

**Our contributions.** We propose WIND a probabilistic foundation model of the atmosphere that can solve a large variety of climate and weather specific downstream tasks, eliminating the need for task-specific fine-tuning. WIND can stabilize long rollouts and allows for explicit guidance of the generative process. By explicitly modeling precipitation, we address a critical gap in current baselines.

## 2 BACKGROUND AND RELATED WORK

Numerical Weather Prediction (NWP) was long established as the standard for operational weather forecasting, making predictions based on numerical fluid dynamics simulations. However, NWP remains computationally extremely demanding and slow Bauer et al. (2015). Recently data-driven deep learning approaches have emerged as a powerful alternative. By leveraging massive amounts of high-dimensional historical data, these models directly try to learn atmospheric dynamics. They shift the heavy computational burden almost entirely to the training phase, enabling real-time inference that is orders of magnitude faster and more efficient than traditional NWP systems. AI-based forecasting models have rapidly evolved over the last years, closing the performance gap with tra-

ditional physical solvers while offering significantly reduced inference costs Price et al. (2025); Chantry et al. (2025); Lam et al. (2023); Bi et al. (2023).

**Diffusion models for spatiotemporal data**. While initial deterministic AI models Bi et al. (2023); Lam et al. (2023) dominated the field, they produce blurry predictions at long lead times as a result of minimizing the mean squared error (MSE) during training Price et al. (2025). Stochastic diffusion models (see Appendix B.1) solve this by modeling the full probability distribution of high-dimensional data Ho et al. (2020); Song et al. (2020). This allows them to generate sharp, coherent structures that reflect the multi-scale nature of the data. Their inherent stochasticity allows them to generate ensemble forecasts, which is essential for quantifying uncertainty and assessing the probability of extreme weather events Li et al. (2023). While diffusion models have revolutionized static image synthesis, adapting them to spatiotemporal dynamics requires specific architectural strategies. We frame atmospheric modeling as a video generation task, where atmospheric variables correspond to channels (like RGB in a video) and time steps correspond to video frames (Figure 1). Currently, autoregressive conditional diffusion models, such as GenCast Price et al. (2025), represent the state-of-the-art in forecasting. They predict the next state given a context history. However, long autoregressive rollouts suffer from error accumulation, often leading to instabilities or divergence Chen et al. (2024). Furthermore, they lack a mechanism to guide the entire sequence toward a global objective, as early predictions are fixed and cannot be retroactively adjusted.

**Autoregressive vs sequence diffusion**. A common alternative in video generation is full-sequence diffusion Ho et al. (2022), which diffuses a fixed number of frames simultaneously with a shared noise level. While this enables globally consistent sampling, these models often struggle with generating long rollouts beyond their training window length. Extending the video autoregressively, by using the last clean frame of the previous window as the first frame of the new window, fails because it creates an out-of-distribution state, as the model requires all frames to share the same noise level. Recent efforts like Elucidated Rolling Diffusion Models (ERDM) Cachay et al. (2025); Ruhe et al. (2024) attempt to mitigate this failure by defining a fixed noise schedule where frames become progressively noisier, effectively encoding the growing uncertainty at longer lead times. However, this rigid structure restricts the model's flexibility during inference.

**Unified modeling with diffusion forcing**. In this work, we leverage diffusion forcing Chen et al. (2024) to overcome the downsides of autoregressive and full-sequence diffusion models. Unlike standard approaches, we train on sequences where each frame is assigned a random, independent noise level. This allows the model to accept clean frames (zero noise) from previous windows as context without distribution shift, enabling stable, arbitrarily long rollouts Chen et al. (2024). This flexible training objective natively supports rolling diffusion-like inference Ruhe et al. (2024); Cachay et al. (2025) without requiring rigid, pre-defined denoising-schedules. This strategy aligns with the reconstruction paradigm, one of the two leading frameworks in Self-Supervised Learning (SSL) Van Assel et al. (2025). One can interpret diffusion noise as a continuous masking mechanism: clean frames serve as unmasked context, while noisy frames are partially masked Hu & Ommer (2024). Diffusion forcing thus generalizes the discrete masking objective, enabling the model to learn robust spatiotemporal representations by reconstructing atmospheric dynamics from varying degrees of corruption.

**Guidance for diffusion models**. A critical advantage of diffusion models over autoregressive approaches is the ability to guide the sampling of a full sequence to minimize a global objective. In image generation, guidance is typically achieved via Classifier-Free Guidance (CFG) Ho & Salimans (2022). CFG offers a tradeoff between diversity and sample quality by jointly training an unconditional and a conditional diffusion model. However, extending CFG to temporal sequences is challenging, as it typically requires specific training strategies such as history dropout to learn an unconditional prior. Recent work on history-guided video diffusion Song et al. (2025) overcomes this limitation by leveraging the unique properties of diffusion forcing. Because the model is trained with independent noise levels, it can naturally estimate both conditional (clean history) and unconditional (fully noisy history) scores, enabling CFG at inference time without training a separate classifier. While history guidance targets temporal consistency, our work aims to enforce physical consistency via external constraints. We frame diverse climate and weather tasks as inverse problems, using posterior sampling to guide the generation process. Unlike standard diffusion posterior sampling Chung et al. (2022), which relies on point estimates, we employ MMPS Rozet et al. (2024).

## 3  APPROACH

The WIND framework decouples learning of the atmospheric dynamics from specific downstream applications. Our pipeline consists of two stages: (1) pre-training a spatiotemporal backbone using diffusion forcing with independent noise levels to learn a flexible generative prior and (2) applying moment matching posterior sampling at inference to solve downstream tasks as inverse problems.

**Problem formulation and notation**.  We denote the atmospheric state as a spatiotemporal tensor $\mathbf{X} = \{\mathbf{x}^1, \dots, \mathbf{x}^T\} \in \mathbb{R}^{T \times C \times H \times W}$, where $T$ is the sequence length (number of frames), $C$ is the number of variables (channels) and $H, W$ represent the spatial resolution. We pre-train the models with sequences of length $T = 5$, with a 6-hours stride, covering $C = 70$ variables at $1.5°$ resolution ($H = 121$ and $W = 240$) (see Appendix A.1).

**Diffusion forcing training**.  Standard video diffusion adds noise to all frames at the same rate Ho et al. (2022). In contrast, following the diffusion forcing paradigm Chen et al. (2024), we sample a *noise level* $k^t \in [0, 1]$ independently for each frame $t \in \{1, \dots, T\}$. This allows the model to learn to predict any frame given any arbitrary combination of clean or noisy context frames. For a given frame $\mathbf{x}^t$ with a noise level $k^t$, the forward diffusion process is given by:

$$\mathbf{z}^t = \alpha(k^t)\mathbf{x}^t + \beta(k^t)\boldsymbol{\epsilon}^t, \quad \boldsymbol{\epsilon}^t \sim \mathcal{N}(\mathbf{0}, \mathbf{I}), \tag{1}$$

where $\alpha(k^t)$ and $\beta(k^t)$ are the signal and noise schedule coefficients defined by the diffusion time $k^t$. The noised sequence is denoted as $\mathbf{Z} = \{\mathbf{z}^1, \dots, \mathbf{z}^T\}$.

The model approximates the score function $\nabla_{\mathbf{Z}} \log p(\mathbf{Z})$ with a neural network $\mathbf{s}_\theta(\mathbf{Z})$ learned through denoising score matching Vincent (2011); Hyvärinen & Dayan (2005). A critical distinction in our approach, motivated by findings from Sun et al. (2025a), is that we **do not** condition the network on the noise levels $k^t$. This forces the model to infer the noise levels (and thus the uncertainty) solely from the input state $\mathbf{Z}$, preventing over-reliance on explicit noise conditions. For more details see Appendix B.3.

**Inference**.  We formulate the inverse problem as recovering $\mathbf{X}$ from observations $\mathbf{Y}$, modeled as:

$$\mathbf{Y} = \mathcal{A}(\mathbf{X}) + \eta, \quad \eta \sim \mathcal{N}(\mathbf{0}, \delta^2\mathbf{I}), \tag{2}$$

where $\mathcal{A}$ is the task-specific forward operator and $\eta$ represents Gaussian measurement noise with variance $\delta^2$. In the case of spatial downscaling (super-resolution) Aich et al. (2026); Hess et al. (2025), the objective is to reconstruct the high-resolution state $\mathbf{X}$ that remains consistent with the observation, given a low-resolution observation $\mathbf{Y}$. Therefore, the task is framed as sampling from the posterior distribution $p(\mathbf{Z}|\mathbf{Y})$.

Using Bayes' rule, the posterior score is decomposed into a prior score and a likelihood score:

$$\nabla_{\mathbf{Z}} \log p(\mathbf{Z}|\mathbf{Y}) = \underbrace{\nabla_{\mathbf{Z}} \log p(\mathbf{Z})}_{\text{Prior Score}} + \underbrace{\nabla_{\mathbf{Z}} \log p(\mathbf{Y}|\mathbf{Z})}_{\text{Likelihood Score}}. \tag{3}$$

The prior score is provided by our trained diffusion model $\mathbf{s}_\theta(\mathbf{Z})$. To estimate the likelihood score $\nabla_{\mathbf{Z}} \log p(\mathbf{Y}|\mathbf{Z})$ we use MMPS Rozet et al. (2024). For more details we refer to Appendix B.1.

**Task formulation**.  We define the $\mathcal{A}$ operator for the various downstream tasks as follows:

- **Probabilistic forecasting:** No guidance. Predict $\mathbf{x}^t$, $t = 2, \dots, T$, given $\mathbf{z}^1 = \mathbf{x}^1$.
- **Spatial downscaling:** Recover high-resolution details from coarse inputs via $\mathcal{A}(\mathbf{X}) = \text{AvgPool}_{s \times s}(\mathbf{X})$.
- **Temporal downscaling:** Recover sub-daily dynamics from daily means via $\mathcal{A}(\mathbf{X}) = \frac{1}{T}\sum_{t=1}^{T}\mathbf{x}^t$.
- **Sparse reconstruction:** Reconstruct the full state from a binary sensor mask $\mathbf{M}$ via $\mathcal{A}(\mathbf{X}) = \mathbf{M} \odot \mathbf{X}$.
- **Global dry air mass conservation:** Enforce constant dry air mass via $\mathcal{A}(\mathbf{X}) = \mathrm{f}_{DAM}(\mathbf{x}^t) = C_{\text{DAM}}$.
- **Generating weather in a warmer climate:** Assume a pseudo-global warming state given by a mean shift on channels $c$ via $\mathcal{A}(\mathbf{X})_c = \left(\frac{1}{HW}\sum_{h,w}\mathbf{x}^t_{c,h,w}\right)$.

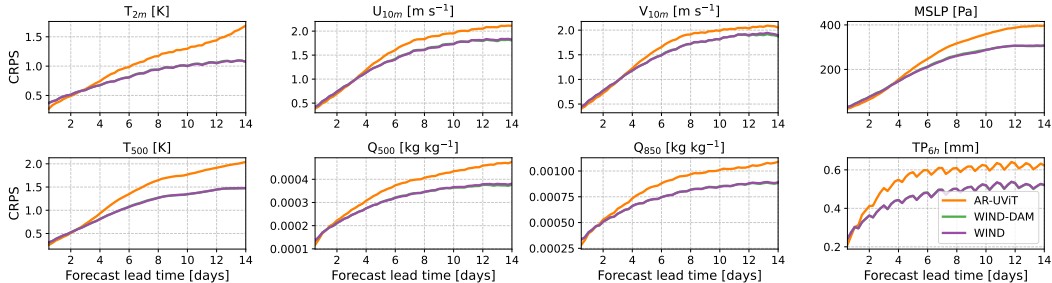

Figure 2: **Probabilistic forecast performance** We evaluate the 14-day forecast skill using the CRPS (lower is better) averaged over 100 initializations in 2021. We compare the unconstrained WIND baseline against the performance with enforced dry air mass conservation. The large overlap demonstrates that the physics constraint does not degrade the probabilistic forecast skill. WIND outperforms the autoregressive AR-UViT baseline after very few days.

## 4 EXPERIMENTS

**Probabilistic forecasting**. Probabilistic forecasting is a fundamental benchmark to validate that WIND has learned a robust, physically consistent prior of atmospheric dynamics. We frame forecasting as a conditional generation task: given a context window of past clean frames $\mathbf{X}_{\text{past}}$, we sample the future frames $\mathbf{X}_{\text{future}}$ by initializing them as pure noise and denoising them using our model. Unlike autoregressive baselines (e.g. GenCast, GraphCast) that optimize specifically for forecasting, our independent noise training formulation allows us to perform forecasting purely at inference time without task-specific fine-tuning.

We evaluate forecast skill on 24 initial conditions from 2021, generating 10-member ensembles for a 14-day lead time. We assess performance using the continuous ranked probability score (CRPS) for accuracy and the spread-skill ratio (SSR) for calibration (see Appendix C.1). Our autoregressive AR-UViT baseline replicates the GenCast forecasting setting C.1.

Figure 2 shows the CRPS for key atmospheric variables. WIND consistently outperforms the autoregressive AR-UViT baseline in CRPS after the first few days, demonstrating superior stability over longer horizons. In terms of calibration, we analyze the spread-skill ratio (SSR) in Figure 7. Ideally, the SSR should be close to 1.0, indicating that the ensemble spread accurately reflects the forecast error. We observe that WIND approaches an SSR of 1.0 over time, transitioning from an initially over-confident state to smoothly saturate at the climatological variance after two weeks as fundamental predictability limits are reached Bauer et al. (2015). Crucially, our method avoids the instability of the AR-UViT baseline, which is also initially under-confident but then tends to overshoot (SSR $>1$) for moisture-related variables (e.g. $Q500$, $Q850$ and $TP6h$).

To evaluate long-term stability, we conduct a 20-year unconstrained rollout initialized in the year 2000. We compare the physical consistency of WIND against an autoregressive diffusion-based baseline, AR-UViT5. Both share the same architecture and sequence length $T$. AR-UViT5 is trained as full sequence diffusion: it receives a clean initial frame as context and is optimized to denoise subsequent frames with a uniform noise level, see Appendix C.1. Figure 20 shows that AR-UViT5 exhibits unphysical spikes across all variables, while WIND maintains physical consistency across the entire spectral range.

**Spatial downscaling**. Spatial downscaling (increasing the resolution) is critical for impact modeling, as climate model projections have coarse resolutions due to computational constraints. While standard reanalysis products like ERA5 are available at 0.25°, long-term climate projections (e.g. CMIP6) are limited to much coarser grids, which smooth out extreme events. Bridging this resolution gap is essential for local risk assessment.

We frame this as an inverse problem, aiming to recover a high-resolution sequence $\mathbf{X} \in \mathbb{R}^{T \times C \times H \times W}$ from a coarsened observation $\mathbf{Y} \in \mathbb{R}^{T \times C \times (H/s) \times (W/s)}$ generated by a pooling operator $\mathcal{A}(\mathbf{X}) = \text{AvgPool}_{s \times s}(\mathbf{X})$. We benchmark WIND against two specialized deterministic models, a Fourier Neural Operator (FNO) (Li et al., 2021) and a UViT Hoogeboom et al. (2023) based model

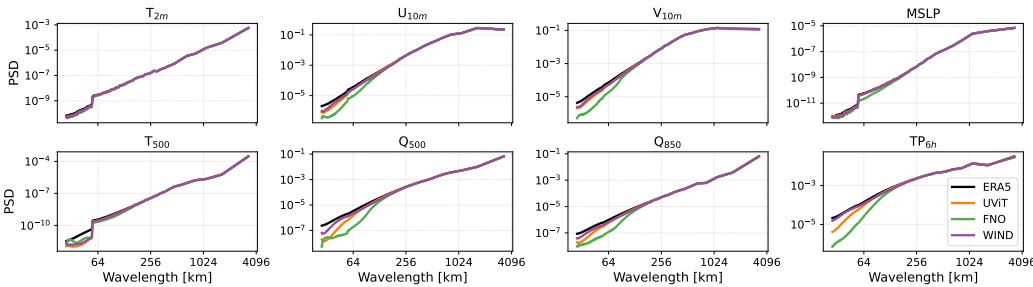

Figure 3: **Power spectra for spatial downscaling.** We compare the PSD of the ERA5 ground truth, a specialized FNO and UViT model, and WIND. WIND closely tracks the energy spectrum of ERA5 across all scales, preserving high-frequency details. In contrast, the deterministic FNO baseline exhibit spectral drop-off at high frequencies. While UViT performs on par with our method for surface variables, it struggles with the atmospheric variables $Q$ and $TP$.

on a $s = 4$ downscaling task for the entire year of 2021 (see Appendix A.2). In principle, we can downscale data from any resolution to match the resolution of our training data.

Downscaling must recover the small-scale physics which is absent in the low-resolution input. As shown in Figure 3, WIND demonstrates superior capability in reconstructing these high-frequency details. The power spectral density (PSD) plots reveal that our model maintains energy levels consistent with the ERA5 ground truth even at the smallest wavelengths, outperforming our UViT baseline. In contrast, the deterministic FNO baseline suffers from spectral bias, exhibiting a sharp drop-off at high frequencies which results in overly smooth, blurry predictions, a known limitation of regression-based objectives. We provide a quantitative comparison in Table 2 and Figure 10. The specialized UViT baseline achieves the lowest RMSE across all variables. This is expected, as deterministic models optimized for pixel-wise error inherently induce spatial smoothing under uncertainty, artificially inflating skill at the expense of physical realism. In contrast, WIND is a generative model that maintains the high-frequencies and structural fidelity of the atmosphere (as evidenced by the PSD). This small-scale intermittency leads to the double-penalty effect during pixel-wise evaluation, where slight spatial displacements of realistic features are penalized more heavily (as both a miss and a false alarm) than heavily smoothed predictions ((Subich et al., 2025)). Generally, it is expected that a model with the same capacity can perform better when fine-tuned to a specific task, especially in the case of downscaling, where the model can learn to only model the high-frequency patterns during training and rely on the low-resolution condition. However, despite not being trained for this task, WIND remains competitive with (and often outperforms) the deterministic FNO baseline in terms of RMSE. While the specialized UViT baseline achieves slightly lower pixel-wise RMSE, our model outperforms it in capturing high-frequency variability for key atmospheric variables like for example specific humidity.

**Temporal downscaling**. Although Earth system models (ESMs) run at fine temporal scales, outputs are frequently aggregated and archived at daily or monthly resolutions to manage data volumes. This temporal aggregation is insufficient for accurate impact modeling, as averaging obscures the sub-daily dynamics of critical extreme events. Despite its importance, temporal downscaling remains largely unexplored compared to its spatial counterpart, with only recent works leveraging generative models to bridge this gap Bassetti et al. (2024); Schmidt et al. (2025). We address this by recovering high-frequency sequences $\mathbf{X} \in \mathbb{R}^{T \times C \times H \times W}$ from temporally aggregated observations $\mathbf{Y} \in \mathbb{R}^{1 \times C \times H \times W}$ (e.g. daily means). We solve temporal downscaling as a pure inference problem via MMPS guidance, ensuring the generated sequence remains consistent with the daily average while generating plausible sub-daily dynamics (see Appendix C.3). We compare our approach against a specialized model trained specifically for temporal downscaling. We use a specialized UViT trained to downscale the daily average field $\mathbf{Y} \in \mathbb{R}^{1 \times C \times H \times W}$ into four 6 hourly frames $\mathbf{X} \in \mathbb{R}^{4 \times C \times H \times W}$.

The primary goal of temporal downscaling is to generate plausible sub-daily dynamics that are absent in the coarser temporally average resolution. As shown in Figure 12, WIND excels at this task. The power spectral density plots show that our model recovers the full energy spectrum of

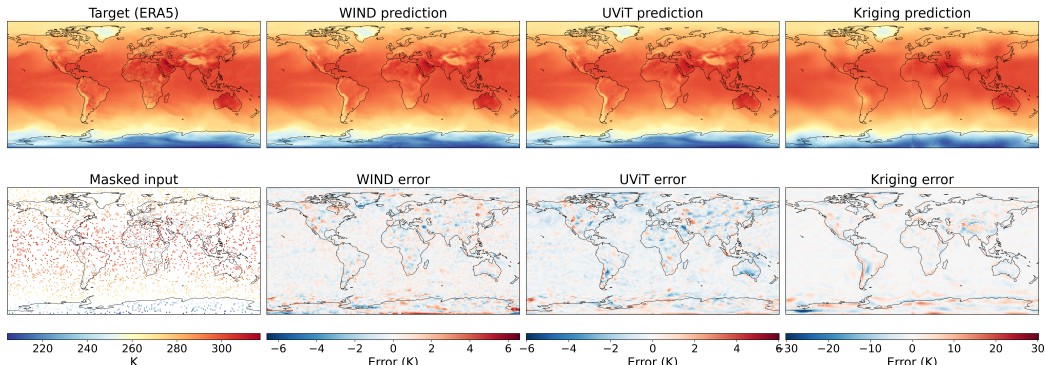

Figure 4: **Qualitative comparison of sparse reconstruction. Left:** ERA5 ground truth (2m Temperature) and **10%** sparse input mask. **Center:** Predictions from WIND, UViT, and Kriging. **Right:** Prediction error relative to ground truth. While WIND and UViT recover physically coherent fields with realistic gradients, Kriging yields overly smooth interpolations that miss fine-grained patterns.

the 6-hourly data, matching the ERA5 ground truth across all wavelengths. This is important for variables like precipitation and surface wind, where the daily mean largely smooths out the variance. For total precipitation our model outperforms the specialized UViT baseline. The histograms further confirm that WIND and the UViT baseline correctly reproduce the sub-daily distributions. This includes the heavy tails of extreme events, which are essential for risk assessment but typically lost in temporally aggregated data. Quantitatively, the specialized UViT baseline achieves lower RMSE, as shown in Table 3 and Figure 13. Analogous to the spatial case, slight temporal phase shifts are heavily penalized by pixel-wise metrics like RMSE, even when the generated weather is physically valid. Figure 14 compares WIND and UViT qualitatively; both models perform on par. To further validate physical fidelity, Figure 11 decomposes the diurnal cycle into harmonic amplitude and phase. WIND demonstrates near-zero amplitude bias and accurate phase locking. This confirms that, despite higher RMSE, the model captures the thermodynamics and thermal inertia correctly.

**Sparse reconstruction**. Global atmospheric reanalysis datasets combine simulations with sparse observations from satellites, weather balloons, and ground stations using data assimilation techniques Hersbach et al. (2020). We address the challenge of reconstructing full global fields from spatially disjoint measurements, a critical task for historical reanalysis and gap-filling satellite data.

We frame this again as an inverse problem: recovering the full state $\mathbf{X}$ from sparse observations $\mathbf{Y}$ defined by a binary masking operator $\mathcal{A}(\mathbf{X}) = \mathbf{M} \odot \mathbf{X}$. Unlike statistical interpolation methods (e.g., Kriging) or specialized models, MMPS allows us to handle arbitrary sensor configurations without retraining (see Appendix C.4). While traditional statistical methods like Kriging minimize error by regressing to the mean, resulting in overly smooth fields, WIND preserves the high-frequency power spectrum of the atmosphere, generating realistic textures even in unobserved regions. Due to the prohibitive $O(N^3)$ computational scaling of Kriging with respect to the number of observation points, we limit its evaluation to a representative day rather than the full evaluation year. As a minimum-variance estimator, Kriging is theoretically guaranteed to produce smoothed fields, making it inherently unsuitable for recovering high-frequency dynamics.

As shown in Table 4, as well as Figure 18 and Figure 19, WIND outperforms the specialized UViT baseline at reconstruction accuracy for the majority of atmospheric variables. Our model excels at predicting large-scale dynamical fields, delivering a substantial reduction in RMSE for metrics such as geopotential and MSLP. This is in contrast to the downscaling tasks, where the specialized model had lower overall RMSE. Here, the specialized UViT struggles to generalize from the extremely sparse 1% input. In contrast, WIND leverages its prior of the atmosphere to fill in 99% unobserved regions coherently, demonstrating that a strong foundation model can outperform specialized training in data-scarce regimes.

The visual comparison in Figure 4 (or in Figure 15 with 1% sparsity) highlights the structural advantages of generative models for the task. Kriging indeed produces an overly smooth field that miss high-frequency weather dynamics.

Visually WIND and UViT both generate sharp, realistic looking fields that are indistinguishable from the ERA5 ground truth. The spectral analysis (Figure 16), confirms that WIND closely tracks the energy spectrum of the ground truth across all scales, whereas UViT suffers from smoothing at high frequencies for some fields like precipitation or specific humidity. Both methods closely follow the ERA5 ground truth distribution (Figure 16 and Figure 17) for most of the variables. However, WIND outperforms the baseline for total precipitation, better covering the extremes events.

**Enforcing global dry air mass conservation**.

Purely data-driven AI forecasting models tend to become unstable or drift into nonphysical states during long rollouts Chattopadhyay et al. (2023). While hybrid architectures like NeuralGCM Kochkov et al. (2024) try to mitigate this long-term instability by coupling a physics-based dynamical core with neural network parameterizations, other recent AI-based approaches rely on external corrective schemes to explicitly enforce the conservation of global energy, moisture budget and dry air mass Sha et al. (2025). We demonstrate that WIND can maintain physical consistency and enforce the global conservation of dry air mass purely at inference time, treating it strictly as an inverse problem (see Appendix C.5). We define a global operator $A_{DAM}$ that computes the global integral of dry air mass (DAM). Using MMPS guidance, we constrain the generation process to satisfy $A_{DAM}(X) = C_{\text{DAM}}$ at every step (see Equation 19).

We evaluate the stability of our method on a 4-year rollout. As shown in Figure 5, a standard free run eventually drifts after 200 days, after initially mimicking the unconstrained fluctuations present in the ERA5 training data. On the other hand, the MMPS-guided run strictly maintains the global DAM at the target value for the entire duration, demonstrating that our framework can correct nonphysical drifts for long rollouts without retraining. Importantly, Figure 2 confirms that the global DAM conservation does not degrade short-term forecast skill across 100 initializations in 2021. While precipitation skill does not improve, this aligns with Sha et al. (2025), who attributes such improvements to moisture and energy constraints. Enforcing these constraints requires surface flux variables that are missing from our dataset.

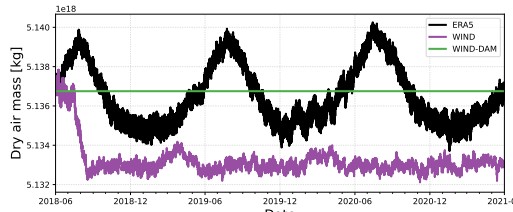

Figure 5: **Long-term stability of dry air mass.** The DAM of WIND without constraint drifts around 200 forecast days. ERA5 ground truth shows a seasonal cycle. WIND with DAM guidance strictly enforces conservation for the entire 4-year rollout.

**Applying the model in a different climate**.

Evaluating global, data-driven AI models under out-of-distribution (OOD) climate scenarios is a major challenge. When initialized in a warmer state, unconstrained AI models inherently drift back to their training climatology during autoregressive rollouts, dissipating the prescribed thermodynamic anomalies Rackow et al. (2024). To overcome this unphysical drift, we propose a generative surrogate scenario to stress-test the model's learned prior. We perturb the initial conditions and use the $\mathcal{A}$ operator to continuously constrain only the global mean thermodynamic state. Crucially, by constraining only the global mean rather than individual grid cells, the model retains the spatial freedom to naturally simulate localized weather dynamics.

We apply this to Storm Bernd (July 2021, Germany), assuming a counterfactual +2K global warming and +14% specific humidity increase via Clausius-Clapeyron scaling Trenberth et al. (2003). While explicitly prescribing this constant offset alters the global thermodynamic response, it acts as a necessary form of thermodynamic conditioning, to sustain the forcing within the AI model. Although this uniform perturbation is a simplified alternative to spatially varying climate change deltas derived from Earth system models Duan et al. (2025), it effectively establishes a warmer background climate and prevents the model from reverting to its training distribution. This enables us to evaluate how the model's learned prior internally resolves localized extreme weather responses under OOD conditions.

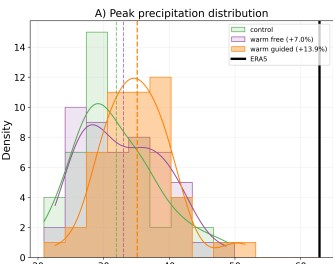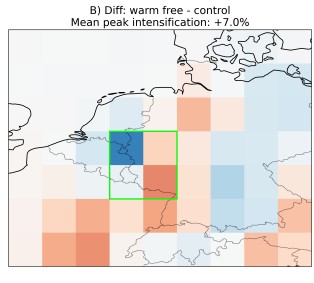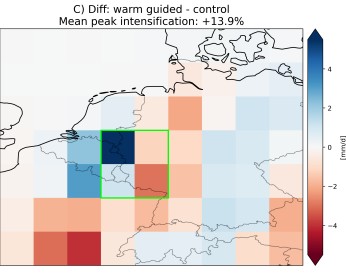

Figure 6: **Ablation of thermodynamic guidance for storm Bernd.** **(A)** Peak 24-hour precipitation density ($N = 50$) within the target region (green box). Dashed lines denote ensemble means; solid black is ERA5. **(B-C)** Spatial difference maps relative to control. The unconstrained warm free run **(B)** dissipates the thermodynamic signal, whereas active guidance **(C)** sustains robust intensification.

To isolate the impact of our guidance mechanism, we conduct an ablation study with three scenarios (see Appendix C.6). We run 24-hours forecasts (4 frames) comparing (i) a **control** run starting from the unperturbed initial condition, (ii) a **warm free** run, where only the initial state is perturbed (warmer and wetter) but the model evolves unconstrained and (iii) a **warm guided** run, with the perturbed initial conditions and where MMPS actively enforces the mean thermodynamic constraint at every step.

As shown in Figure 6, the warm guided ensemble effectively sustains the out-of-distribution thermodynamic shift. To compute the intensification, we average the pairwise differences of the peak precipitation pixels between the control and the warm guided runs over the target region (49°-52°N, 5.5°-8.5°E). Across the 50 ensemble members, we observe a mean peak intensification of +13.9%, closely matching the theoretical Clausius-Clapeyron baseline expected for a +2K warming (∼14%). Crucially, the unconstrained **warm free** ablation run retained only 50.3% of this signal (+7.0%). Without active guidance, the model's learned prior effectively diffuses the initial state anomaly back toward its training climatology Rackow et al. (2024), leading to a significant underestimation of the extreme event. Meanwhile, our guided framework maintains excellent structural stationarity: for the 500 hPa geopotential height, we observe a structural similarity (SSIM) of $> 0.98$ and zero-pixel displacement of the storm center, while horizontal wind speeds changed by less than 1%.

Finally, while the ensemble mean aligns with theoretical expectations, the spatial response remains highly heterogeneous, which is consistent with observational findings Traxl et al. (2021). Our generative ensemble approach reveals the importance of capturing these localized tail risks. In the worst-case guided realization (Member 48), we observed a local precipitation increase of +56.9% (+18.7 mm/d), whereas the warm free run produced a maximum increase of only +2 mm/d. This behavior aligns with literature finding that localized extreme precipitation often exhibits Super-Clausius-Clapeyron scaling, exceeding the ∼14% baseline due to complex non-linear feedbacks Berg et al. (2013).

## 5 DISCUSSION AND FUTURE WORK

In this work, we presented WIND, a framework that marks a shift from specialized, single-task models to a unified probabilistic foundation model of the atmosphere. Our results highlight an inherent trade-off between task-specific specialization and flexibility. While specialized baselines of comparable size achieve slightly lower RMSE in our experiments, we found that our method often exhibits less spectral smoothing and preserves more high-frequency details for atmospheric variables (Figure 3 and Figure 12). By decoupling atmospheric dynamics from task specific objectives, WIND supports arbitrary inference constraints, such as enforcing physical consistency or reconstructing arbitrary sensor inputs. We even outperform our specialized baselines for sparse reconstruction and forecasting. We argue that this versatility outweighs the marginal pixel-wise gains of single-task models. Diffusion forcing allows WIND to perform long-term rollouts without accumulation of artifacts compared to full sequence diffusion Figure 20.

Our experiments demonstrate that active guidance is essential for out-of-distribution evaluation, as the unconstrained model inherently diffuses thermodynamic anomalies back toward the training distribution Rackow et al. (2024). To counteract this drift, recent AI adaptations have had to rely on explicit dynamic nudging or costly ensemble filtering Duan et al. (2025). In contrast, our framework sustains a prescribed thermodynamic background state to prevent OOD drift, while leaving the local dynamics unconstrained. Conceptually, our thermodynamic experiment shares similarities with the pseudo-global warming (PGW) approach Schär et al. (1996); Brogli et al. (2023), which explores how specific historical events would unfold under altered thermodynamic states Shepherd et al. (2018) driven by climate change. Before our model can be used to answer PGW storyline type questions, further validation is required to ensure that the AI model's unconstrained dynamics respond realistically to the altered thermodynamics. A direction for future work is to systematically evaluate these dynamics, for example, by training the model on pre-industrial control simulations, applying a global warming offset, and comparing the generated dynamics against fully coupled socio-economic pathway (SSP) projections. We also emphasize that this single-event analysis serves as an initial proof of concept. Localized extremes should be interpreted as plausible tail risks of the learned distribution rather than strict causal predictions.

Because AI weather models generally lack physical constraints, they are prone to violating fundamental conservation laws, which in turn drives unphysical drift and error accumulation during autoregressive rollouts Sha et al. (2025). Our results demonstrate that WIND successfully enforces global integral constraints (e.g. dry air mass) purely at inference time. However, we explicitly note that while our framework stabilizes global budgets, it does not currently enforce local conservation laws (such as exact grid-cell level flux balances). Achieving strict local conservation remains a challenge for AI models, representing an important direction for future research.

Our framework aligns with the perfect prognosis paradigm Van Der Meer et al. (2023), making it highly relevant for downscaling Earth system models. While domain shifts, caused by biases in ESMs, typically hinder the direct application of ERA5-trained downscaling models, recent two-stage approaches successfully decouple bias correction from downscaling Aich et al. (2026); Wan et al. (2023). Unlike previous frame-wise downscaling approaches Aich et al. (2026); Hess et al. (2025), we enable the model to capture temporal correlations in high-frequency patterns absent in static frames. Recovering extremes smoothed out in coarse climate projections is critical for impact modelers and policy makers. By training with larger window sizes, WIND can be used to downscale monthly data to daily resolution, effectively recovering extremes.

Despite promising results, our approach has limitations that we leave for future research. The primary drawback is inference speed. The iterative denoising and MMPS gradient calculations result in a high computational cost. This can be addressed via distillation techniques Salimans & Ho (2022); Sabour et al. (2025); Potaptchik et al. (2026). Additionally, while the model captures the relative intensification of events, it underestimates extreme events like Storm Bernd. This is likely a combination of our coarse $1.5°$ spatial resolution and the general difficulty of out-of-distribution generalization of data-driven models Sun et al. (2025b). By training at $0.25°$ resolution, WIND could better resolve these extremes.

Beyond these improvements, we envision several conceptual extensions. To capture longer-range phenomena (e.g. the El Niño Southern Oscillation) and enforce thermodynamic budgets, future iterations could incorporate sea-surface temperatures and energy fluxes. The additional information could improve the model's forecast performance, especially for precipitation Sha et al. (2025). Moreover, we could move beyond reanalysis data by adapting the framework to directly assimilate raw observations from satellites and stations, effectively turning WIND into a real-time data assimilation system Andry et al. (2025); Yang et al. (2025). Finally, by training on large-scale climate simulations with explicit external forcings, future work could explore the large-scale response of atmospheric dynamics to varying greenhouse gas concentrations. Thus, WIND could serve as a computationally efficient emulator for alternative climate scenarios, complementing computationally expensive ESMs.

Ultimately, WIND marks a significant step towards a general-purpose atmospheric model, offering a unified framework that adapts to diverse downstream tasks while respecting the complex physical laws of the earth system.

ACKNOWLEDGMENTS

The ELLIS Unit Linz, the LIT AI Lab, the Institute for Machine Learning, are supported by the Federal State Upper Austria. We thank the projects FWF AIRI FG 9-N (10.55776/FG9), AI4GreenHeatingGrids (FFG- 899943), Stars4Waters (HORIZON-CL6-2021-CLIMATE-01-01), FWF Bilateral Artificial Intelligence (10.55776/COE12). We thank NXAI GmbH, Audi AG, Silicon Austria Labs (SAL), Merck Healthcare KGaA, GLS (Univ. Waterloo), TÜV Holding GmbH, Software Competence Center Hagenberg GmbH, dSPACE GmbH, TRUMPF SE + Co. KG.

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

# A EXPERIMENTAL DETAILS

## A.1 DATASET

**ERA5**.    We use the ERA5-dataset Hersbach et al. (2020), a state-of-the-art reanalysis dataset provided by the European Center for Medium-Range Weather Forecasting (ECMWF). We use the 1.5° resolution of the data (with a grid size of 240 ×121 pixel) at 6 hourly temporal resolution provided by Weatherbench-2 Rasp et al. (2024). We include 70 prognostic variables (input and output): **Surface Variables** (5): Precipitation (p), 2m temperature (2t), mean sea level pressure (mslp), and the u and v components of 10m wind (10u, 10v). **Pressure Level Variables** (65): Temperature (t), geopotential (z), specific humidity (q), and the u and v components of wind (u, v) across 13 distinct pressure levels.

**Pre-processing**.    We normalize each variable, at each pressure level to zero mean and unit variance using statistics calculated over the training set. For precipitation variables, which exhibit heavy-tailed distributions with values spanning multiple orders of magnitude, we apply a log-transformation prior to the standard normalization following Aich et al. (2026); Hess et al. (2025). Specifically, we map the raw precipitation values $x$ (in m) to $x = \log_{10}(1000x + 1)$. The factor 1000 converts the values to millimeters, and the constant 1 ensures numerical stability for zero-precipitation regions. We further incorporate auxiliary static and dynamic features into the input state. Static features include the land-sea mask, soil type, and surface geopotential. The surface geopotential is normalized to zero mean and unit variance, while the masks are kept as binary identifiers. To preserve the spherical geometry of the Earth without discontinuities at the date line, we embed the spatial grid coordinates $(\phi, \lambda)$ into 3D Cartesian space as $(\sin \phi, \cos \phi, \cos \phi \sin \lambda)$ . Furthermore, to explicitly encode the temporal cycle, we append sine and cosine embeddings of both the annual cycle (year progress) and the diurnal cycle (local time of day, derived from UTC time and longitude) to the input channels

## A.2 ARCHITECTURE & HYPERPARAMETERS

**Architecture**.    For our model and all UViT baselines we use the same UViT Hoogeboom et al. (2024) backbone and use adaptions of the Transformer layers as done in Song et al. (2025) to adapt the model to temporal data. For the downstream tasks we modified the training paradigm (deterministic / diffusion) and training setup (input and outputs) when necessary.

We configure the UViT with a hierarchical structure across four spatial resolutions. Starting from a base channel dimension of 256, the channels progress as (256, 512, 1024, 2048) across resolutions. After initial patchification with patch size 2, we employ a hybrid block design to balance computational efficiency with global context modeling. Specifically, we use four residual blocks (He et al., 2016) at each of the two highest spatial resolutions, and four Transformer blocks (Vaswani et al., 2017) at each of the two lowest resolutions. Each Transformer block employs 4 attention heads and 1D rotary positional embeddings (RoPE) (Su et al., 2021) applied across both spatial and temporal dimensions. This configuration results in a total parameter count of approximately 458 million. We utilize a temporal window of $T = 5$ in order to predict a full day at 6 hourly resolution given one state. Each timestep comprisies 70 atmospheric variables at a resolution of $240 \times 121$ pixels.

For the downscaling we additionally use a FNO (Li et al., 2021) as an additional baseline. We use $L = 8$ layers, a hidden width of 128, and a grid positional embedding, and we truncate spectral convolutions at $k_{max} = [64, 128]$ modes. This results in a total parameter count of approximately 545 million.

**Hyperparameters**.    We use a cosine learning rate schedule with a linear warmup period of 5 epochs, reaching a peak learning rate of $1 \times 10^{-4}$ before decaying to a minimum of $1 \times 10^{-6}$. The model is trained for a maximum of 60 epochs with a global effective batch size of 16. We use the Adam, clipping gradients above a norm of 0.8. To improve training efficiency, we utilize bfloat16 mixed precision. During evaluation, we use an EMA version of the model, with an EMA decay rate of 0.999. Since the native ERA5 spatial grid of 240×121 (longitude × latitude) does not result in an even number after multiple downsampling stages, we bilinearly interpolate the input to a 240×128 grid before the network and interpolate it back to the original resolution after the final layer.

During inference, all our diffusion based models use DDIM sampling with a 15 deterministic steps (thus $\eta = 0$). When using MMPS for guidance, we use 2 iteration steps for the conjugate gradient method and a noise variance $\delta^2 = 0.0015$.

## B APPROACH DETAILS

### B.1 DIFFUSION MODELS BACKGROUND

Diffusion models define a stochastic *forward* process that corrupts clean data $\mathbf{x} \in \mathbb{R}^D$ by gradually adding noise over a continuous time interval $k \in [0, 1]$. This process can be modeled as a Stochastic Differential Equation (SDE) (Song et al., 2020):

$$d\mathbf{z}_k = \mathbf{f}(\mathbf{z}_k, k)dk + g(k)d\mathbf{w} \tag{4}$$

where $\mathbf{w}$ is a standard Wiener process. The drift $\mathbf{f}(\mathbf{z}_k, k)$ and diffusion coefficient $g(k)$ are chosen such that $\mathbf{z}_0 = \mathbf{x}$ (clean data) and $\mathbf{z}_1$ approaches a standard Gaussian distribution.

To generate samples, we simulate the *reverse* process (Anderson, 1982) which runs backward in time from $k = 1$ to $k = 0$:

$$d\mathbf{z}_k = [\mathbf{f}(\mathbf{z}_k, k) - g(k)^2 \nabla_{\mathbf{z}_k} \log p_k(\mathbf{z}_k)]dk + g(k)d\tilde{\mathbf{w}} \tag{5}$$

where $d\tilde{\mathbf{w}}$ is a reverse-time Wiener process. Since the true score function $\nabla_{\mathbf{z}_k} \log p_k(\mathbf{z}_k)$ is intractable, it is approximated by a neural network $\mathbf{s}_\theta(\mathbf{z}_k, k) \approx \nabla_{\mathbf{z}_k} \log p_k(\mathbf{z}_k)$, trained via denoising score matching (Song et al., 2020; Vincent, 2011).

### B.2 DIFFUSION FORCING TRAINING DETAILS

Standard video diffusion adds noise to all frames at the same rate Ho et al. (2022). In contrast, following the diffusion forcing paradigm Chen et al. (2024), we sample a *noise level* $k \in [0, 1]$ independently for each frame $t \in \{1, \ldots, T\}$. This allows the model to learn to predict any frame given any arbitrary combination of clean or noisy context frames. For a given frame $\mathbf{x}^t$, let $k^t$ be its sampled noise level. The forward diffusion process is given by:

$$\mathbf{z}^t = \alpha(k^t)\mathbf{x}^t + \beta(k^t)\boldsymbol{\epsilon}^t, \quad \boldsymbol{\epsilon}^t \sim \mathcal{N}(\mathbf{0}, \mathbf{I}), \tag{6}$$

where $\alpha(k^t)$ and $\beta(k^t)$ are the signal and noise schedule coefficients defined by the diffusion time $k^t$. The noised sequence is denoted as $\mathbf{Z} = \{\mathbf{z}^1, \ldots, \mathbf{z}^T\}$.

**Objective**. The neural network $\hat{\mathbf{X}}_\theta(\mathbf{Z})$ is trained to predict the clean atmospheric state $\mathbf{X} = \{\mathbf{x}^1, \ldots, \mathbf{x}^T\}$ given the noised sequence $\mathbf{Z}$. A critical distinction in our approach, motivated by findings from Sun et al. (2025a), is that we **do not** condition the network on the noise levels defined by the set of indices $\mathbf{k} = \{k^1, \ldots, k^T\}$. The model is strictly a mapping $\hat{\mathbf{X}}_\theta : \mathbb{R}^{T \times C \times H \times W} \to \mathbb{R}^{T \times C \times H \times W}$. The objective function is the weighted mean squared error,

$$\mathbb{E}_{\mathbf{X}, \boldsymbol{\epsilon}, \mathbf{k}} \left[ \sum_{t=1}^{T} \sum_{c=1}^{C} w_c \sum_{h,w=1}^{H,W} a_{h,w} \|\mathbf{x}_{c,h,w}^t - \hat{\mathbf{X}}_\theta(\mathbf{Z})_{c,h,w}^t\|_2^2 \right], \tag{7}$$

where $t$ indexes the frame index, $c$ the physical variables (channels) and $(h, w)$ the spatial coordinates. To account for varying grid cell sizes, $a_{h,w}$ represents the normalized cell area such that $\frac{1}{HW} \sum_{h,w} a_{h,w} = 1$. The channel-specific weights $w_c$ are adopted from previous work in atmospheric modeling (Lam et al., 2022; Price et al., 2023). We can infer the score function $\mathbf{s}_\theta(\mathbf{Z})$ from the data prediction model $\hat{\mathbf{X}}_\theta(\mathbf{Z})$ (Kingma & Gao, 2023) using:

$$\mathbf{s}_\theta(\mathbf{Z})^t = -\beta(k^t)^{-2} \left( \mathbf{z}^t - \alpha(k^t)\hat{\mathbf{X}}_\theta(\mathbf{Z})^t \right), \tag{8}$$

where $\mathbf{s}_\theta(\mathbf{Z}) = \{\mathbf{s}_\theta(\mathbf{Z})^1, \ldots, \mathbf{s}_\theta(\mathbf{Z})^T\}$.

We use a rectified noise schedule (Liu et al., 2023; Lipman et al., 2023), that means our coefficients $\alpha(k^t) = k^t\alpha_{\min} + (1 - k^t)$ and $\beta(k^t) = k^t + (1 - k^t)\beta_{\min}$, with $\alpha_{\min} = \beta_{\min} = 0.001$. We sample each $k^t \sim \mathcal{U}(0, 1)$ independently.

Table 1: Wall-clock runtime and peak memory for unguided and DAM-guided (MMPS) inference on the forecasting task.

| Run | DDIM Steps | CG Steps | Runtime [s] | Cost $\times$ vs Unguided | Peak Memory [GB] |
|---|---|---|---|---|---|
| Unguided | 15 | - | 0.5573 | 1.00 | 9.59 |
| Unguided | 30 | - | 1.1133 | 1.00 | 9.59 |
| Guided | 15 | 1 | 3.1930 | 5.74 | 14.82 |
| Guided | 15 | 2 | 4.5189 | 8.12 | 14.82 |
| Guided | 15 | 3 | 5.8466 | 10.51 | 14.82 |

## B.3 SAMPLING DETAILS

For our forecasting tasks, we provide the model with one clean state and denoise all the others with the same noise level $k$. For all other tasks, we initialize all states noisy and denoise, and just use guidance to reach the desired output.

For frame denoising, we employ DDIM sampling. Specifically, to transition from noise level $k$ to $k'$ ($k' < k$), we denoise each frame within the window via:

$$\mathbf{z}^t \leftarrow \alpha(k')\hat{\mathbf{X}}_\theta(\mathbf{Z})^t + \beta(k')\sqrt{1 - \eta\tau}\frac{\mathbf{z}^t - \alpha(k)\hat{\mathbf{X}}_\theta(\mathbf{Z})^t}{\beta(k)}$$
$$+ \beta(k')\sqrt{\eta\tau}\epsilon, \tag{9}$$

with $\epsilon \sim \mathcal{N}(\mathbf{0}, \mathbf{I})$ and

$$\tau = 1 - \frac{\alpha(k)^2\beta(k')^2}{\alpha(k')^2\beta(k)^2}.$$

Under guidance, we additionally update each $\mathbf{z}^t$ using the likelihood score $\nabla_\mathbf{Z} \log p(\mathbf{Y}|\mathbf{Z})$, where the likelihood is approximated as:

$$p(\mathbf{Y}|\mathbf{Z}) \approx \mathcal{N}(\mathcal{A}(\hat{\mathbf{X}}_\theta(\mathbf{Z})^t), \mathbf{\Sigma}(k)). \tag{10}$$

Several methods can be used to estimate the covariance $\mathbf{\Sigma}(k)$. Diffusion posterior sampling (Chung et al., 2022), for example, adopts the measurement noise covariance, which corresponds to $\mathbf{\Sigma}(k) = \delta^2\mathbf{I}$ in our setting. In this work, we employ moment matching posterior sampling (MMPS) (Rozet et al., 2024), which estimates $\mathbf{\Sigma}(k)$ via the conjugate gradient method. For further details see Rozet et al. (2024).

## B.4 COMPUTATIONAL COST

Table 1 reports wall-clock runtime, cost relative to unguided sampling, and peak memory usage, averaged over 10 runs after 3 burn-in runs on a NVIDIA RTX PRO 6000. We benchmark the forecasting task, where we generate 4 future frames conditioned on 1 clean context frame. Guided runs additionally enforce dry air mass (DAM) conservation via MMPS at every denoising step. Unguided sampling scales approximately linearly with DDIM steps, running in $0.56\,\mathrm{s}$ for 15 steps and $1.11\,\mathrm{s}$ for 30 steps, with a peak memory footprint of $9.6\,\mathrm{GB}$ in both cases. Guided sampling via MMPS introduces additional cost due to the conjugate gradient (CG) solve at each denoising step: with 15 DDIM steps, runtime scales approximately linearly with CG iterations, from $3.2\,\mathrm{s}$ (1 CG step, $5.7\times$ overhead) to $5.8\,\mathrm{s}$ (3 CG steps, $10.5\times$ overhead), with peak memory increasing to $14.8\,\mathrm{GB}$.

## C DOWNSTREAM TASK DETAILS

### C.1 FORECASTING

**Baselines**.    To simulate the GenCast architecture, we train our model with a sequence length of $T = 3$. In this configuration, we condition on two clean frames and denoise only the final frame, setting $\mathbf{k} = \{0, 0, k\}$. This effectively models the transition probability $p(\mathbf{x}^t|\mathbf{x}^{t-2}, \mathbf{x}^{t-1})$.

Additionally, to evaluate the advantages of diffusion forcing for long-range rollouts, we also trained a variant called UViT-5 ($T = 5$). In this setup, we provide a single clean frame and simultaneously denoise the subsequent four frames at a uniform noise level, such that $\mathbf{k} = \{0, k, k, k, k\}$.

**Evaluation metrics**.   Due to the highly chaotic dynamics of the atmosphere, probabilistic forecasts accompanied by uncertainty quantification are key for evaluating weather forecast models. The spread-skill ratio (SSR) Fortin et al. (2014) and continuous ranked probability score (CRPS) Gneiting & Raftery (2007) are the standard ensemble metrics to assess the forecast performance. The SSR tests the models calibration by comparing the ensemble's spread (standard deviation) to the root-mean-square error of the ensemble mean, where a ratio close to 1 indicates that the forecast uncertainty accurately reflects the actual error. The spread is defined as:

$$\text{Spread} = \sqrt{\frac{1}{HW} \sum_{h,w} a_{h,w} \frac{1}{M-1} \sum_{m=1}^{M} (\hat{\mathbf{x}}_{m,h,w} - \bar{\mathbf{x}}_{h,w})^2}, \tag{11}$$

with the ensemble mean of the predictions

$$\bar{\mathbf{x}}_{h,w} = \frac{1}{M} \sum_{m=1}^{M} \hat{\mathbf{x}}_{m,h,w}. \tag{12}$$

The skill is defined as:

$$\text{Skill} = \sqrt{\frac{1}{HW} \sum_{h,w} a_{h,w} (\bar{\mathbf{x}}_{h,w} - \mathbf{x}_{h,w})^2}. \tag{13}$$

Combining both results in the SSR:

$$\text{SSR} = \sqrt{\frac{M+1}{M}} \frac{\text{Spread}}{\text{Skill}}. \tag{14}$$

The CRPS measures the overall accuracy of a probabilistic forecast by quantifying the integrated squared difference between the forecast's cumulative distribution function and the observed step function, penalizing both bias and lack of sharpness. The CRPS acts as a mean absolute error for probabilistic forecasts, measuring the distance between the entire range of predicted forecasts and the single ground truth observation. We use the following CRPS definition:

$$\text{CRPS} = \frac{1}{HW} \sum_{h,w} a_{h,w} \left[ \frac{1}{M} \sum_{m=1}^{M} |\hat{\mathbf{x}}_{m,h,w} - \mathbf{x}_{h,w}| \right.$$
$$\left. - \frac{1}{2M(M-1)} \sum_{m=1}^{M} \sum_{m'=1}^{M} |\hat{\mathbf{x}}_{m,h,w} - \hat{\mathbf{x}}_{m',h,w}| \right]. \tag{15}$$

## C.2  Spatial downscaling

**Formulation**.   We aim to recover a high-resolution sequence $\mathbf{X} \in \mathbb{R}^{T \times C \times H \times W}$ given a low-resolution observation sequence $\mathbf{Y} \in \mathbb{R}^{T \times C \times (H/s) \times (W/s)}$. Here, $T = 5$ denotes the sequence length and $s \in \mathbb{N}$ is the downscaling factor. In our experiment, the target $\mathbf{X}$ consists of $1.5°$ ERA5 fields, while the condition $\mathbf{Y}$ is the corresponding $6°$ low-resolution sequence ($s = 4$). The relationship is defined by the forward operator $\mathcal{A}$ applied frame-wise for $t = 1 \ldots T$:

$$\mathbf{y}_t = \mathcal{A}(\mathbf{x}_t) = \text{AvgPool}_{s \times s}(\mathbf{x}_t). \tag{16}$$

**Baselines**.   We benchmark our approach against specialized architectures optimized for spatial downscaling. Specifically, we train the FNO and U-ViT described in A.2 as deterministic mappings between the input and target resolutions. To account for the downscaling factor of $s = 4$ while maintaining the models' requirement for consistent input-output dimensions, the low-resolution inputs are projected back to the high-resolution grid using nearest-neighbor interpolation. All baseline models operate on individual frames independently, representing a temporal constraint of $T = 1$.

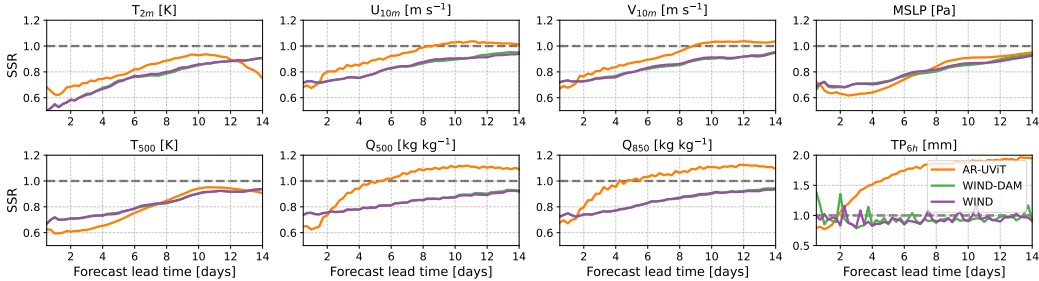

Figure 7: **Spread-skill ratio.** We assess the reliability of the probabilistic forecast using the spread-skill ratio (SSR), where a value of 1.0 (dashed line) indicates perfect calibration. WIND and its physically constrained variant WIND-DAM start slightly over-confident (SSR $< 1$) but steadily approach ideal calibration over the 14-day horizon without overshooting. In contrast, the autoregressive AR-UViT baseline exhibits rapidly increasing spread, drifting into under-confidence (SSR $> 1$) for variables like specific humidity ($Q_{500}$) and total precipitation ($TP_{6h}$). The significant overlap between the purple and green lines confirms that enforcing the dry air mass constraint preserves the probabilistic calibration of the ensemble.

**Extended results**.    In addition to the power spectrum shown in the main text, the histograms in Figure 9 confirm that WIND preserves the statistical properties of the atmosphere. For heavy-tailed variables like 6-hour precipitation ($TP_{6h}$) and specific humidity ($Q_{500}$ and $Q_{850}$), our model successfully reproduces the distribution of the high-resolution data, whereas deterministic baselines often under-predict extremes.

We also tested how well the predictions align with the low-resolution condition. After re-applying the operator $\mathcal{A}$, we found a pearson correlation of 0.96 for WIND, matching the specialized baselines. Table 2 provides the full RMSE breakdown while Figure 10 compares the normalized RMSEs of our baselines and WIND. While pixel-wise metrics favor the specialized UViT, WIND remains competitive with the FNO baseline despite not being trained on the task,

Table 2: **Quantitative comparison for spatial downscaling.** We report the absolute RMSE averaged over all pressure levels. While the specialized UViT baseline achieves the lowest RMSE, WIND outperforms the specialized FNO without any task-specific training.

| Variable | WIND | UViT | FNO |
|---|---|---|---|
| Temperature (3D) | 0.63 | **0.47** | 0.66 |
| Geopotential (3D) | 45.17 | **25.86** | 52.44 |
| Specific humidity (3D) | 0.0005 | 0.0005 | 0.0005 |
| U-Wind (3D) | 1.89 | **1.47** | 2.08 |
| V-Wind (3D) | 1.76 | **1.38** | 2.01 |
| 2m temperature | 0.76 | **0.57** | 0.70 |
| MSLP | 42.68 | **30.58** | 48.53 |
| 10m U-Wind | 0.93 | **0.76** | 1.02 |
| 10m V-Wind | 0.95 | **0.76** | 1.04 |
| Precipitation | 1.77 | **1.43** | 1.55 |

## C.3    TEMPORAL DOWNSCALING

**Formulation**.     Temporal downscaling aims to recover a high-frequency sequence $\mathbf{X} \in \mathbb{R}^{T \times C \times H \times W}$ given a temporally aggregated observation $\mathbf{Y} \in \mathbb{R}^{1 \times C \times H \times W}$ (e.g. a daily mean). For our 6-hourly data, $T = 4$ corresponds to the snapshots within a 24-hour window. The relationship is defined by the forward operator $\mathcal{A}$, which aggregates frames over the temporal dimension,

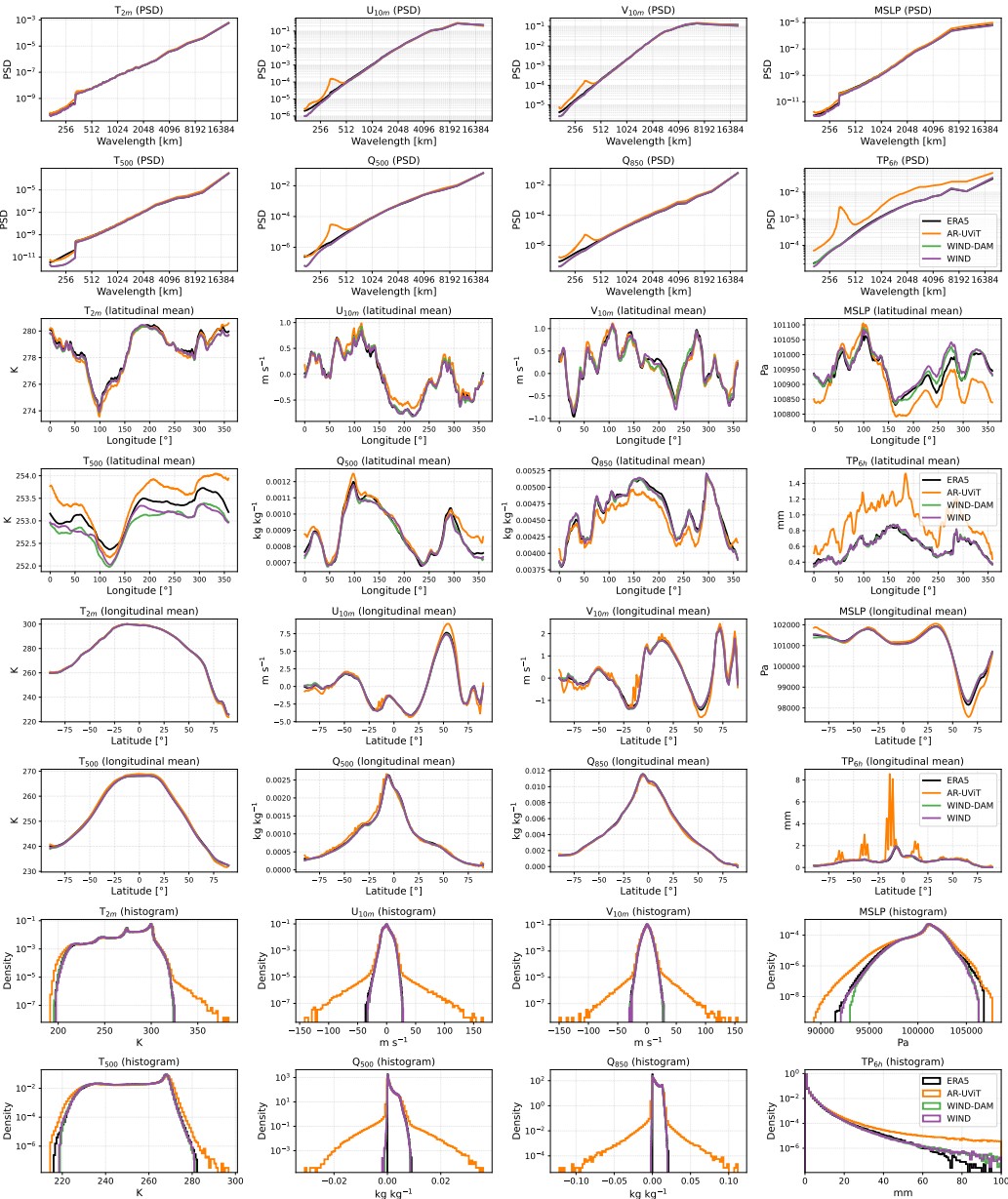

Figure 8: **Spectrum and distribution for forecasting.** Top rows: WIND accurately models the forecast spectrum; in contrast, AR-UViT struggles with high frequencies and precipitation modeling. Middle rows: AR-UViT overestimates precipitation, with notable artifacts near the equator. Bottom rows: AR-UViT histograms shows increasing values in all fields from repeated autoregressive steps, whereas WIND accurately tracks the ERA5 ground truth.

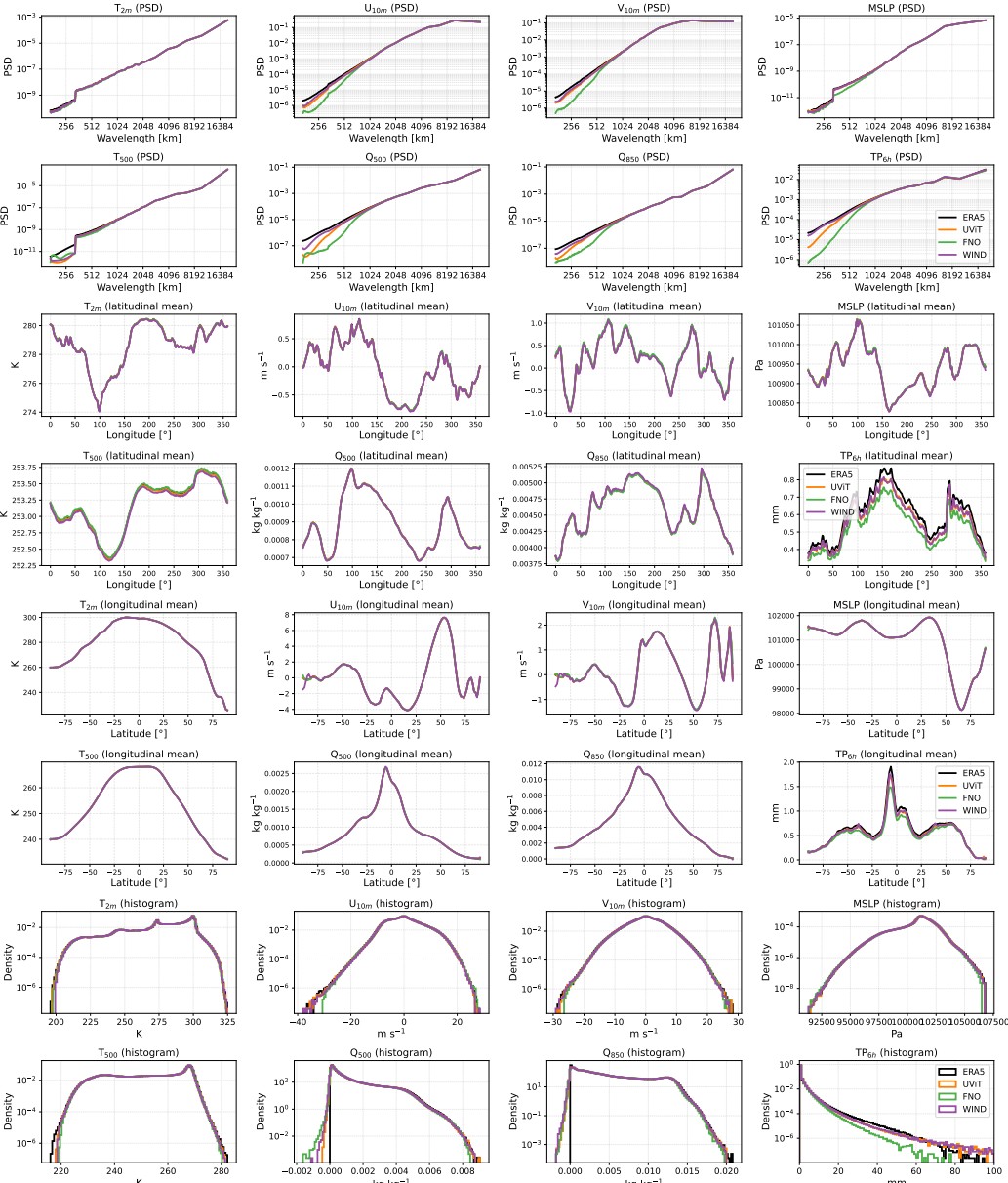

Figure 9: **Spectrum and distribution for spatially downscaled fields.** Top rows: We compare the PSD of the ERA5 ground truth, deterministic FNO, UViT and WIND. WIND closely tracks the energy spectrum of ERA5 across all scales, preserving high-frequency details. In contrast, the deterministic FNO baseline exhibit spectral drop-off at high frequencies. Middle rows: WIND and UViT perform on par at reproducing the latitudinal and longitudinal means. FNO is worse for precipitation. Bottom rows: The histograms confirm that observation.

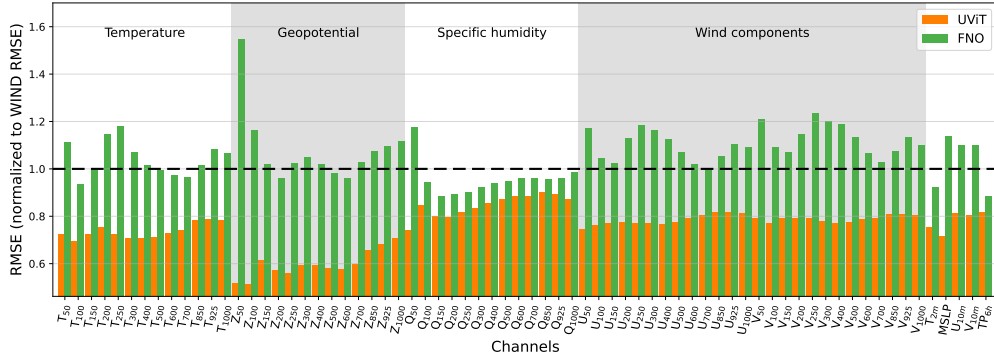

Figure 10: **RMSE comparison of spatial downscaling.** We compare the RMSE of the baselines relative to our method (dashed line at 1.0) to account for different scales in different fields. While the specialized UViT achieves lower RMSE by optimizing for the mean, WIND outperforms the FNO on several variables and remains competitive on others, despite not being trained on the task.

effectively smoothing them:

$$\mathbf{Y} = \mathcal{A}(\mathbf{X}) = \frac{1}{T} \sum_{k=1}^{T} \mathbf{x}_k. \tag{17}$$

Our model is trained on a window of $K = 5$ frames. To perform temporal downscaling, we generate a sequence $\mathbf{X}$ where the first $T = 4$ frames are constrained to match the daily observation $\mathbf{Y}$. The fifth frame is unconstrained, demonstrating the model's flexibility to handle cases where the window size is not perfectly aligned with a downstream task. We sample from the posterior $p_\theta(\mathbf{X}|\mathbf{Y})$ using MMPS guidance, ensuring that the temporal aggregation of the generated sequence is consistent with the daily ground truth. In principle, the target resolution is constrained only by the window size. We evaluate the ability to recover high-frequency patterns lost during the temporal averaging, along side the ability to still align with the daily average field. We temporally downscale every day in 2021 from the ERA5 ground truth and compare to the 6 hourly ground truth.

**Baselines**. For temporal downscaling, we employ a U-ViT baseline with a temporal horizon of $T = 4$. The model receives the daily mean—repeated across all four temporal input slots and is trained via MSE loss to regress the original 6-hourly sequences.

Table 3: **Quantitative comparison for temporal downscaling.** We report the absolute RMSE averaged over all pressure levels for WIND and a task specific diffusion baseline (UViT). The conditional UViT baseline achieves a lower RMSE.

| Variable | WIND | UViT |
|---|---|---|
| Temperature (3D) | 0.79 | **0.55** |
| Geopotential (3D) | 84.47 | **42.06** |
| Specific Humidity (3D) | 0.0004 | **0.0003** |
| U-Wind (3D) | 2.36 | **1.61** |
| V-Wind (3D) | 2.66 | **1.70** |
| 2m Temperature | 0.71 | **0.46** |
| MSLP | 83.91 | **40.98** |
| 10m U-Wind | 1.09 | **0.67** |
| 10m V-Wind | 1.22 | **0.72** |
| Precipitation | 1.68 | **1.10** |

### C.4 SPARSE RECONSTRUCTION

**Background**.

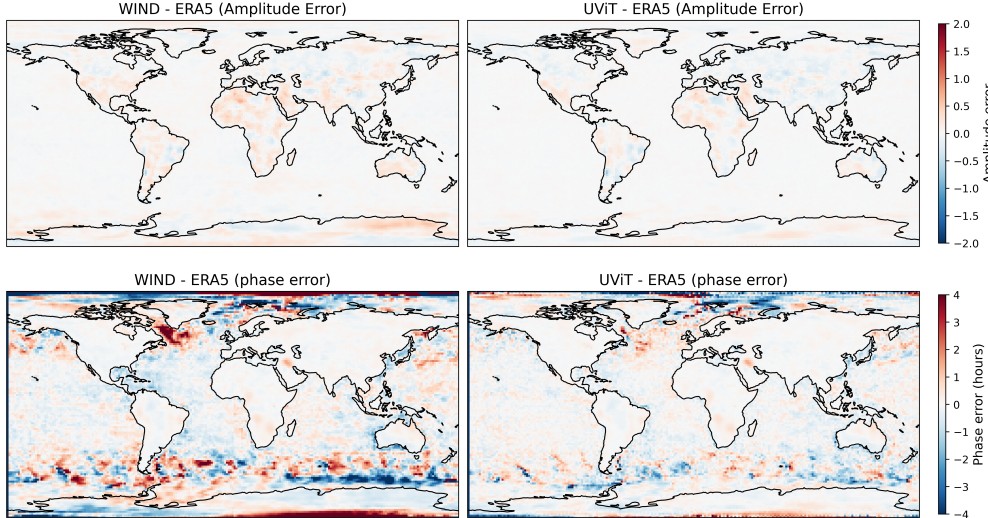

Figure 11: **Validation of sub-daily dynamics via diurnal harmonic analysis.** We evaluate the physical fidelity of the generated 6-hourly fields by decomposing the diurnal cycle into amplitude and phase for 2m Temperature. The plots show the pixel-wise error relative to ERA5 ground truth for WIND and UViT. (Top Row) amplitude bias: the daily mean baseline (right) exhibits a massive negative bias, confirming the significant variance lost by averaging. Both diffusion models (left/middle) exhibit near-zero bias, demonstrating accurate recovery of the diurnal range. (Bottom Row) phase lag: measuring the shift in peak time relative to ERA5. Over land, both models show negligible phase error, correctly capturing the thermal lag of peak temperature. Note that phase noise over oceans is expected due to the negligible diurnal amplitude in those regions.

Satellite measurements are inherently limited by orbital characteristics that create spatial discontinuities. Consequently, the ability to reconstruct global atmospheric states from sparse observations is critical for both modern analysis and extending datasets into historical eras lacking satellite coverage. Diffusion models have emerged as powerful tools for such reconstruction tasks in image processing Saharia et al. (2022), PDEs Amorós-Trepat et al. (2026), and weather Li et al. (2024); Kishikawa et al. (2025). Recently, Li et al. (2024), proposed a specialized framework (S³GM) to perform sparse reconstruction at inference time using a custom conditional SDE solver. In this work, we demonstrate that the physics prior of our pre-trained diffusion model can solve the sparse reconstruction task purely at inference time using MMPS. This approach eliminates the need to train specialized conditional models specifically for sparse reconstruction tasks and complex sampler modifications. We compare our approach against traditional baselines such as Kriging Cressie (1990), a Gaussian Process, based interpolation method widely used in geosciences. Unlike to traditional approaches, MMPS is completely independent of the sensor and does not need to be tuned for different sensor types, compared to traditionally statistical or conditional training based methods.

**Formulation** We frame sparse reconstruction as the recovery of the full global atmospheric state $\mathbf{X}$ from a set of sparse, point-wise observations $\mathbf{Y}$. Referring to the general inverse formulation in Section 3, the forward operator $\mathcal{A}$ for this task is defined as a binary masking operation:

$$\mathbf{Y} = \mathcal{A}(\mathbf{X}) = \mathbf{M} \odot \mathbf{X} \tag{18}$$

where $\mathbf{M}$ is a binary mask representing the spatial locations of sensors (e.g. weather stations or satellite tracks).

**Baselines**. For spatial reconstruction, we train a U-ViT ($T = 1$) to recover the full state $\mathbf{X}$ from sparse observations $\mathbf{Y}$. The model receives the masked observations concatenated with the binary mask as an additional input channel. We employ a dynamic masking strategy, sampling sparsity levels between 1% and 10% with randomized spatial patterns. The architecture is optimized via MSE loss to reconstruct the complete field from these partial observations.

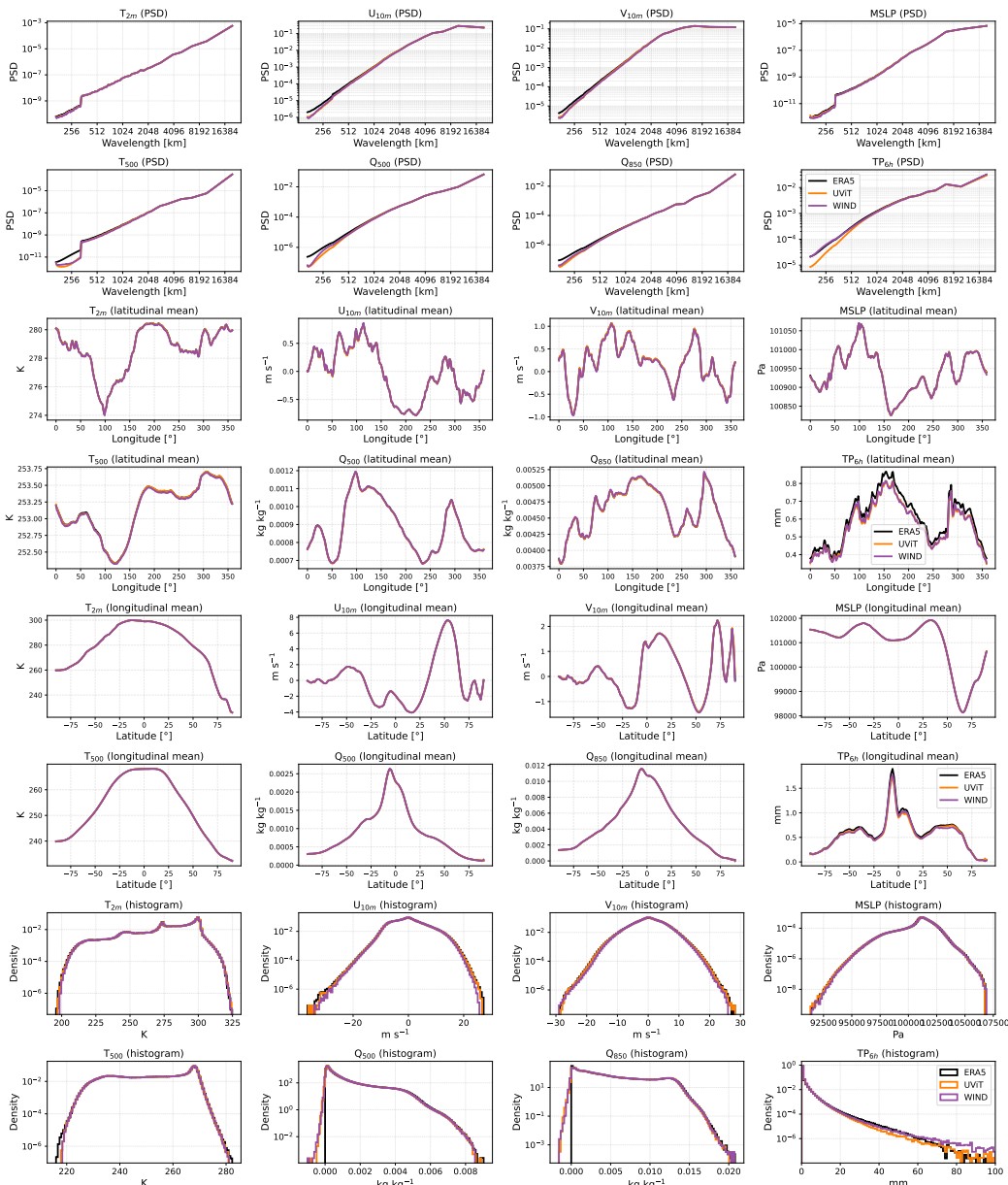

Figure 12: **Spectrum and distribution for temporally downscaled fields.** Top rows: WIND agrees with frequency spectrum of ERA5 extremely well, even for precipitation where UViT struggles. Middle rows: The latitudinal and longitudinal means show that both models agree with ERA5. Bottom rows: The histograms confirm that both models reproduces the probability distributions of ERA5 well.

## C.5 ENFORCING GLOBAL DRY AIR MASS CONSERVATION

**Motivation**. A major limitation of purely data-driven AI forecasting models is that they often become unstable for longer rollouts. While these instabilities come partly from architectural choices, they are also rooted in the inability of AI models to accurately obey the underlying physical conservation laws. To address this, recent literature has tried to enforce physical quantities directly. The two leading approaches are modifying the loss function via soft constraints Verma et al. (2024) or adding specific neural network layers as hard constraints to strictly enforce global conservation of these quantities Sha et al. (2025); Harder et al. (2023); Watt-Meyer et al. (2025). Most AI based

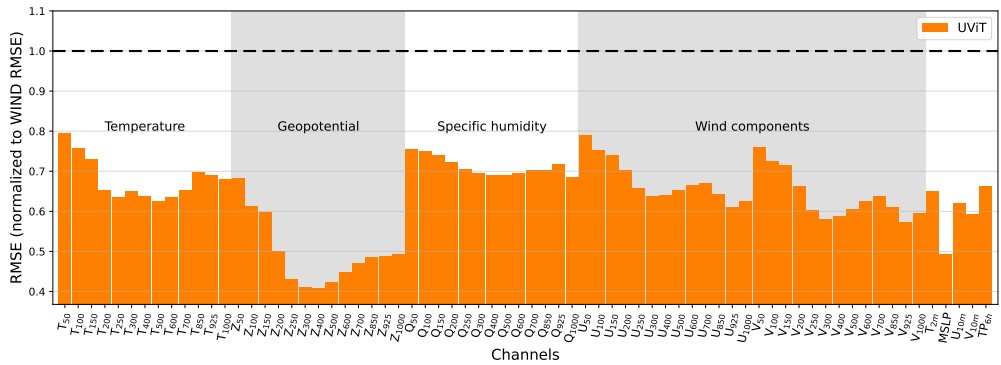

Figure 13: **RMSE comparison of temporal downscaling.** We compare the RMSE of the baselines relative to our method (dashed line at 1.0) to account for different scales in different fields. The specialized UViT achieves lower RMSE than WIND.

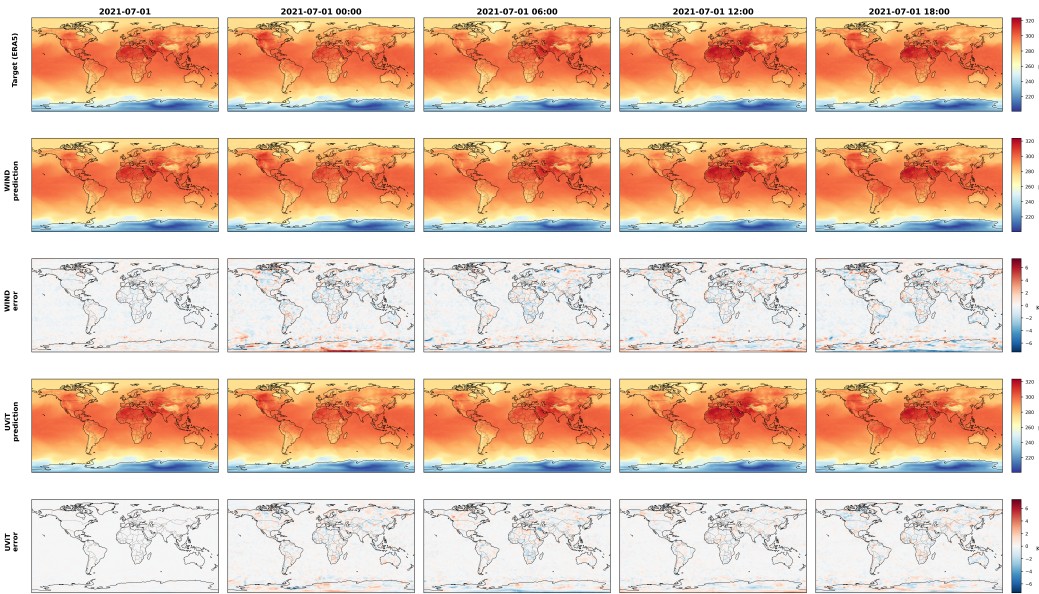

Figure 14: **Qualitative comparison of temporal downscaling for 2m temperature.** The first column displays the daily average, while the subsequent four columns show the 6-hourly high-frequency sequences. The top row shows the ERA5 target on July 1, 2021. The second and fourth rows show predictions from WIND and the specialized UViT baseline, respectively, with their corresponding pixel-wise error maps shown in the third and fifth rows.

weather models do not enforce conservation laws, partly because they are trained on ERA5 reanalysis data Hersbach et al. (2020), which itself does not strictly conserve mass and energy due to the underlying data assimilation process Tootoonchi et al. (2025). Recent work by Sha et al. (2025) demonstrated that enforcing global conservation of energy, moisture budget and DAM improves forecast performance particularly for precipitation, while reducing the drizzle bias (the tendency of AI models to predict light rain everywhere).

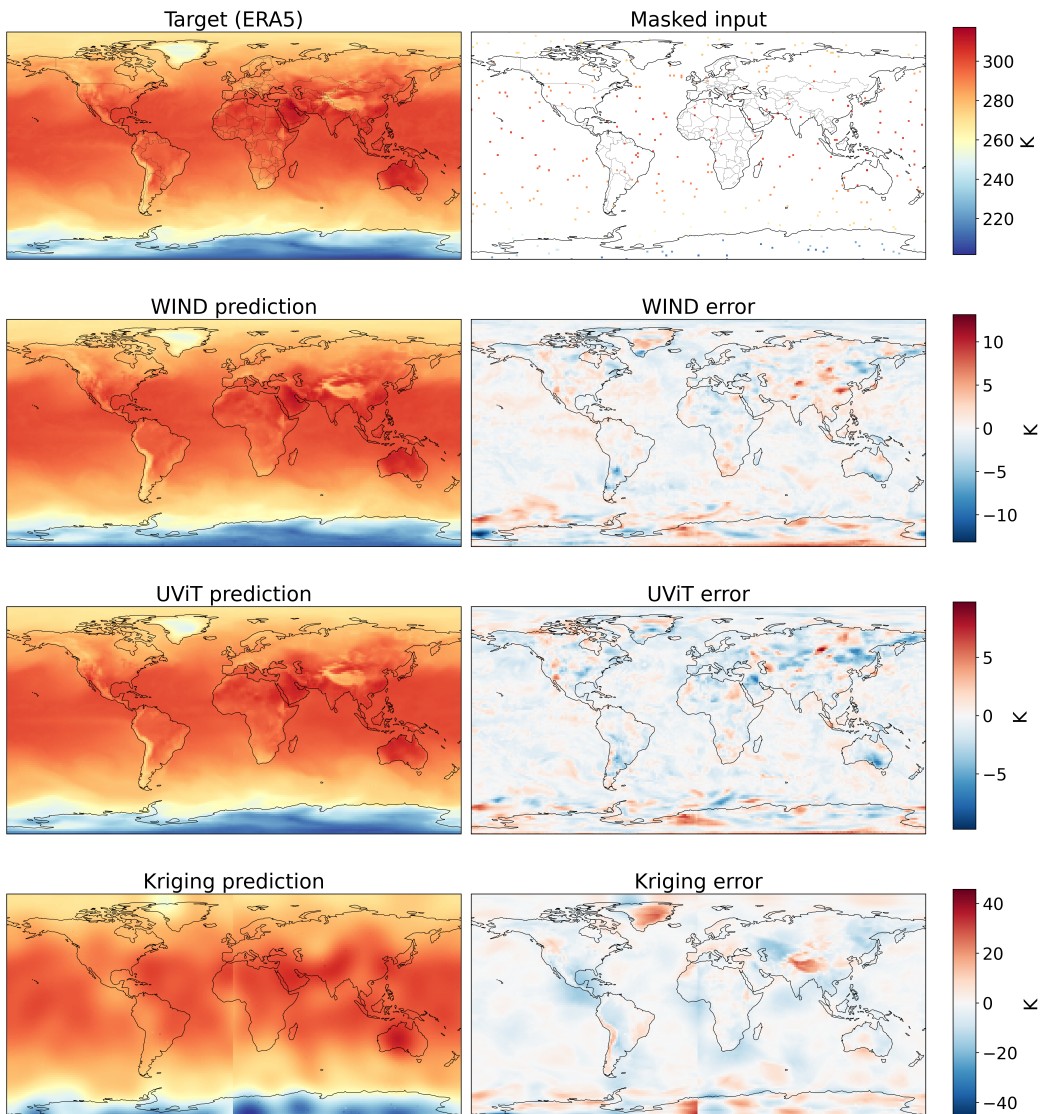

Figure 15: **Qualitative comparison of sparse reconstruction. Left:** ERA5 ground truth (2m Temperature) and **1%** sparse input mask. **Center:** Predictions from WIND, UViT, and Kriging. **Right:** Prediction error relative to ground truth. While WIND and UViT recover physically coherent fields with realistic gradients, Kriging yields overly smooth interpolations that miss fine-grained weather patterns.

**Formulation.** We enforce constant dry air mass via $\mathcal{A}(\mathbf{X}) = \mathrm{f}_{DAM}(\mathbf{x}^t) = C_{\mathrm{DAM}}$. The function $\mathrm{f}_{DAM}$ is derived as:

$$p_{\mathrm{sfc}} = p_{\mathrm{MSLP}} \exp\left(-\frac{\Phi_{\mathrm{sfc}}}{R_d T_{\mathrm{2m}}}\right)$$

$$\mathrm{TWP} = \frac{1}{g} \int_{p_{\mathrm{top}}}^{p_{\mathrm{sfc}}} Q(p)\, dp$$

$$m_{\mathrm{dry}} = \frac{p_{\mathrm{sfc}}}{g} - \mathrm{TWP}$$

$$\mathrm{f}_{\mathrm{DAM}}(\mathbf{x}^t) = \sum_{h,w} a_{h,w} m_{\mathrm{dry}}(h, w) \tag{19}$$

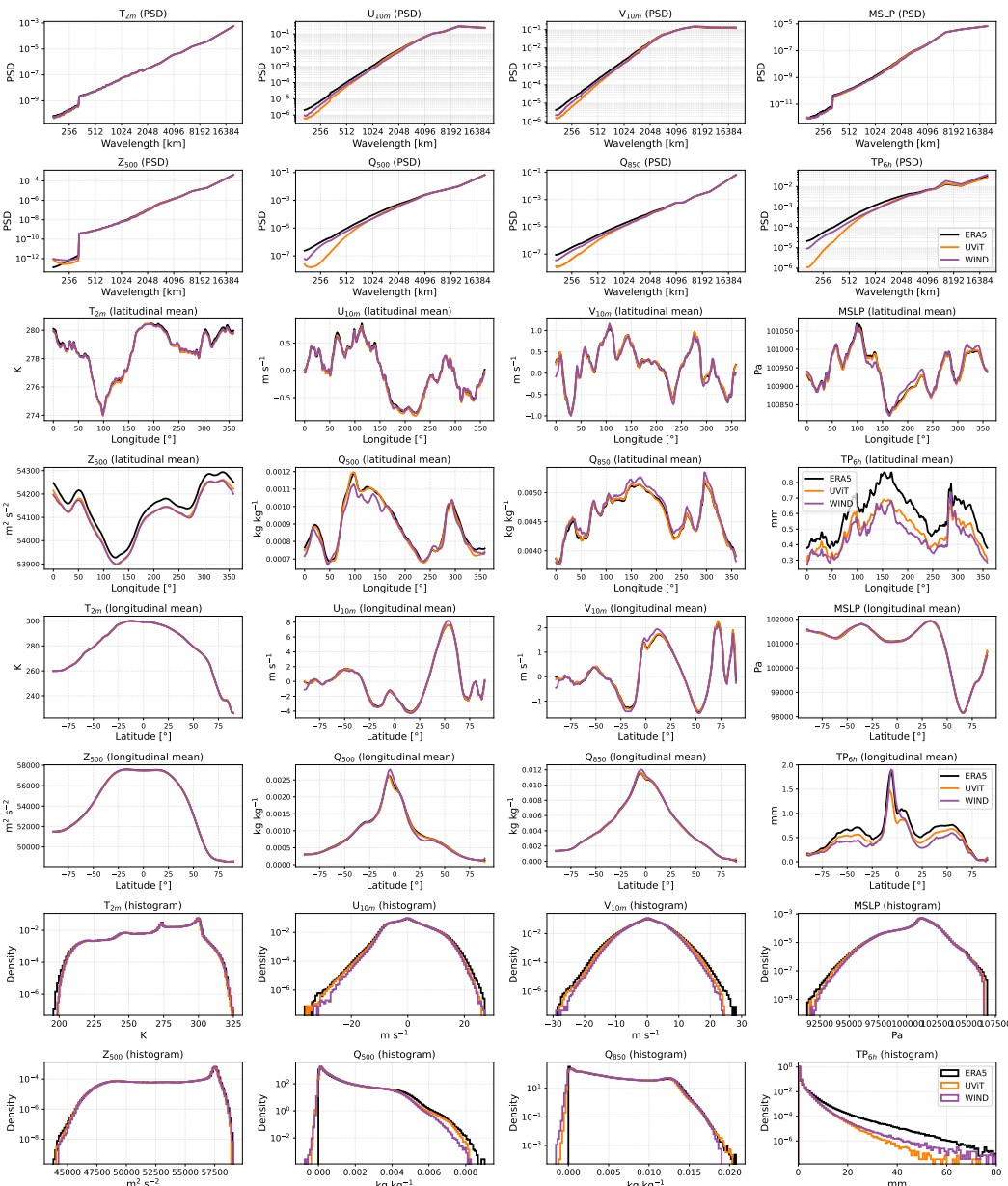

Figure 16: **Spectrum and distribution for sparse reconstruction (1 % sparsity).** Top rows: WIND agrees with frequency spectrum of ERA5, even at high frequencies where UViT struggles. Middle rows: The latitudinal and longitudinal means show that both models are in sync with ERA5, except for precipitation. Bottom rows: The histograms confirm that both models reproduces the full probability distributions. WIND performs slightly better for precipitation.

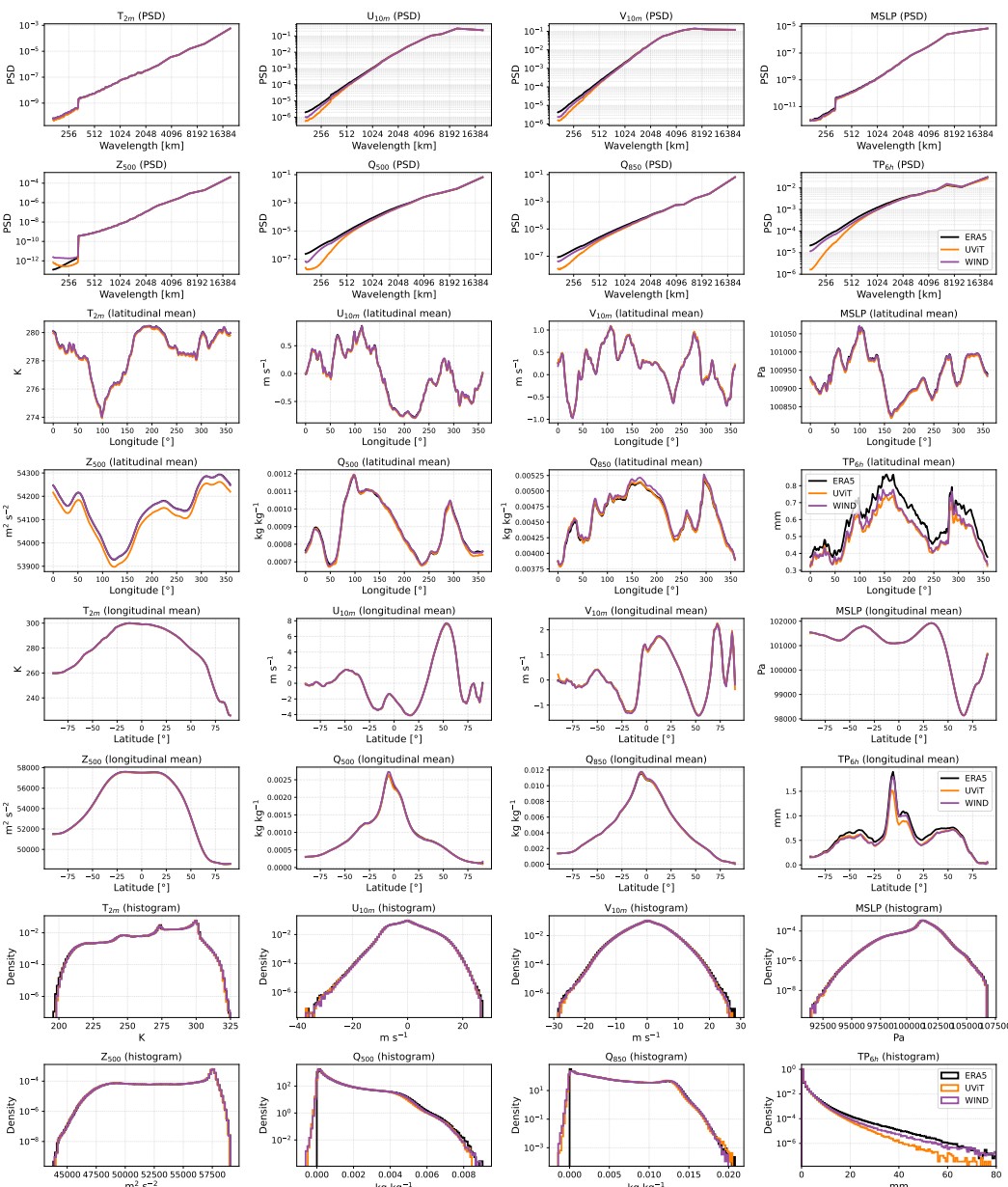

Figure 17: **Spectrum and distribution for sparse reconstruction (10 % sparsity).** Top rows: WIND agrees with frequency spectrum of ERA5, even at high frequencies where UViT struggles. Middle rows: The latitudinal and longitudinal means show that both models align well with ERA5, with slight deviations for precipitation. Bottom rows: The histograms confirm that both models reproduces the full probability distributions. WIND is superior for approximating the histogram for precipitation.

Table 4: **Comparing absolute RMSEs for spatial reconstruction with sparsity** 1%. We compare inference-only method WIND against a Gaussian Process baseline (Kriging) and a specialized conditional diffusion model (UViT). For the atmospheric variables we averaged the RMSE values over all pressure levels. WIND outperforms the baselines on most of the variables.

| Variable | WIND | UViT |
|---|---|---|
| Temperature (3D) | **0.65** | 0.68 |
| Geopotential (3D) | **48.64** | 80.97 |
| Specific Humidity (3D) | **0.0006** | 0.0007 |
| U-Wind (3D) | **1.84** | 1.87 |
| V-Wind (3D) | 1.85 | **1.84** |
| 2m Temperature | **0.83** | 0.86 |
| MSLP | **47.12** | 51.19 |
| 10m U-Wind | **0.95** | 0.99 |
| 10m V-Wind | **1.00** | 1.01 |
| Precipitation | 0.0017 | **0.0016** |

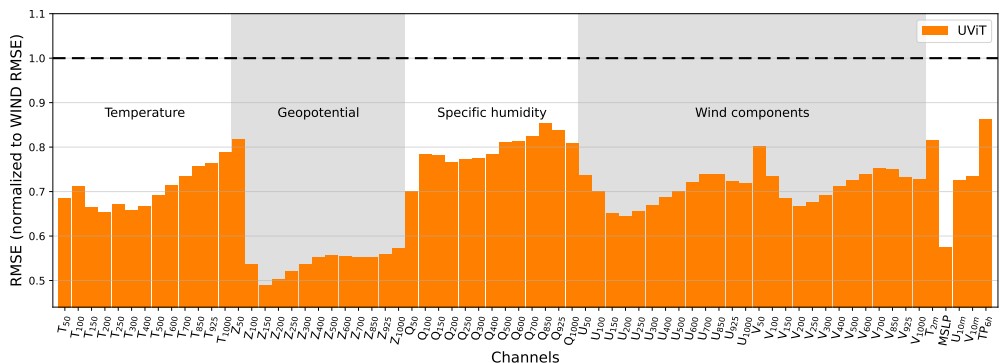

Figure 18: **RMSE comparison for sparse reconstruction (1% sparsity).** We compare the RMSE of the specialized UViT baseline relative to WIND (dashed line at 1.0). WIND outperforms the specialized model (bars > 1.0) on the majority of variables, particularly for fields like geopotential and specific humidity.

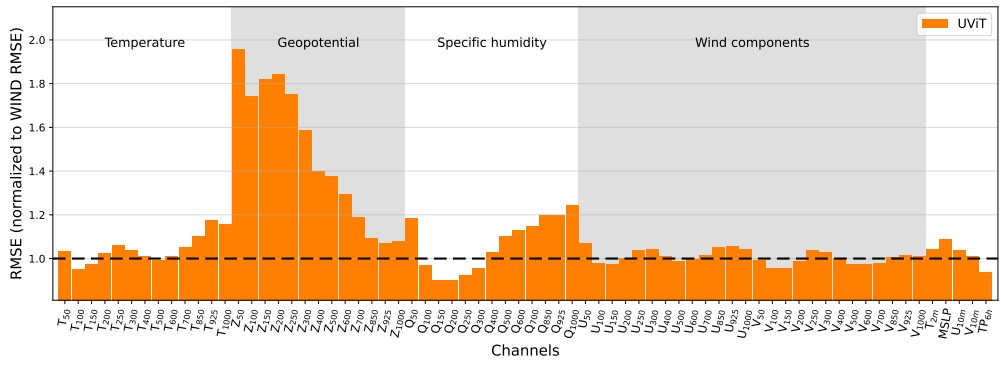

Figure 19: **RMSE comparison for sparse reconstruction (10% sparsity).** We compare the RMSE of the specialized UViT baseline relative to WIND (dashed line at 1.0). WIND outperforms the specialized model (bars > 1.0) on the majority of variables, particularly for fields like geopotential and specific humidity.

where $p_{\text{sfc}}$ is the derived surface pressure and $p_{\text{MSLP}}$ is the mean sea level pressure. The term $\Phi_{\text{sfc}}$ represents the surface geopotential, $R_d$ is the gas constant for dry air, and $T_{2\text{m}}$ is the 2m tempera-

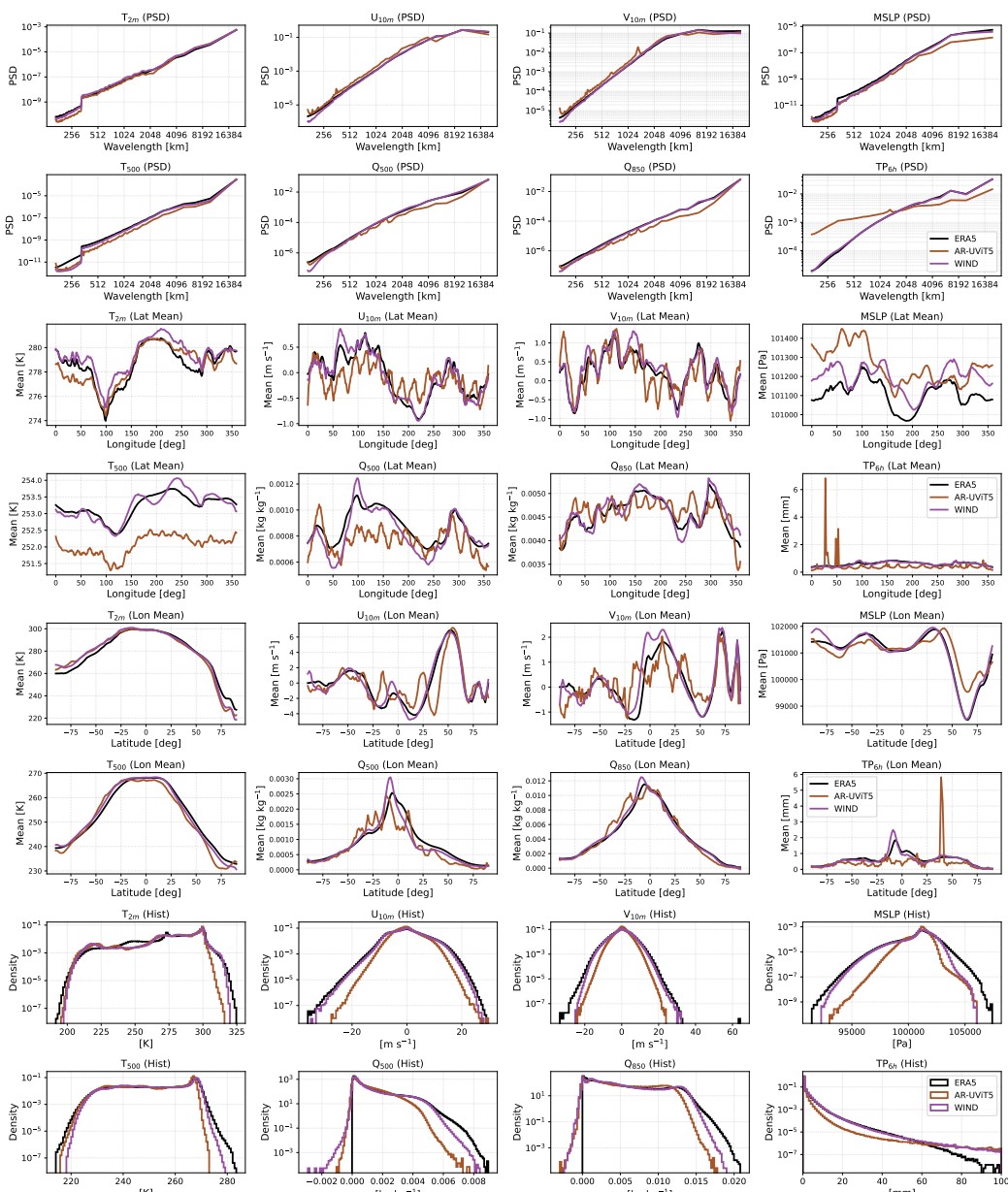

Figure 20: **Comparing long-term rollout stability of diffusion forcing vs full sequence diffusion.** We compare the statistical properties of a 20-year unconstrained forecast generated by our model WIND and the autoregressive baseline AR-UViT5 (full sequence diffusion) against the ground truth ERA5. Top rows: The PSD plots show that AR-UViT5 produces unphysical spikes across all variables, particularly for precipitation. In contrast, WIND accurately preserves energy across the full spectrum. Middle rows: The latitudinal and longitudinal means show that WIND is overall in sync with ERA5, while the baseline exhibits significant biases. Bottom rows: The histograms confirm that WIND faithfully reproduces the full probability distributions only slightly overestimating the tails in precipitation. The baseline collapses completely for some distributions.

ture. For the vertical moisture integration, $g$ denotes gravitational acceleration, $Q(p)$ is the specific humidity at pressure level $p$, and TWP is the total water path. Finally, $a_{h,w}$ represents the area weighting. In Figure 5, we guide the rollout using the numerical DAM value of the first clean frame, denoted as $C_{\text{DAM}}$, which is used to initialize the rollout.

## C.6 GENERATING WEATHER IN A WARMER CLIMATE

We compare an unguided ensemble run starting from the historical initial condition to two warming runs, one guided and one evolving freely. In both warm runs we perturb the initial state by adding +2K to all temperature channels and scaling the specific humidity channels by a factor of $1.07^{\Delta T = 2K}$ (based on Clausius-Clapeyron scaling), leaving dynamic variables unchanged.

In the guided run we enforce the thermodynamic anomaly throughout the generation process by framing it as an inverse problem. We define the $\mathcal{A}$ operator as the spatial average over the temperature and specific humidity fields. The target $\mathbf{y}$ contains the spatial average for all thermodynamic variables for each frame. The average is computed over the historical ground truth. We increase the temperature means $\mathbf{y}_T$ by +2K and increase the specific humidity means $\mathbf{y}_Q$ by +14%. To address the numerical scale disparity between temperature ($\sim 10^2$) and humidity ($\sim 10^{-3}$), we weight the respective channels based on the inverse magnitude of the perturbations to ensure balanced gradient contributions. Inference is performed using 10 steps with $\eta = 1$, 10 conjugate gradient steps and noise variance $\delta^2 = 1e^{-3}$.

