# OpenReview forum: "WIND: Weather Inverse Diffusion for Zero-Shot Atmospheric Modeling"
_ICLR.cc/2026/Workshop/FM4Science — ICLR 2026 Workshop FM4Science Poster_

### Official Review · Reviewer_Kc87 · 2026-02-21
**A diffusion-based atmospheric foundation model by casting it as an inverse problem and solving via moment matching posterior sampling**

**Rating:** 6
**Confidence:** 4

**Review:**

## 1. Summary

This paper proposes WIND, a diffusion-based atmospheric foundation model trained with a self-supervised video reconstruction objective using diffusion forcing (independent noise per frame) and deployed zero-shot on many downstream tasks by casting each as an inverse problem and solving via moment matching posterior sampling (MMPS) at inference. The authors argue that this yields a unified, task-agnostic atmospheric prior that can support probabilistic forecasting, spatial and temporal downscaling, sparse reconstruction, constraint enforcement, e.g., dry air mass conservation, and counterfactual storyline generation without task-specific fine-tuning.

## 2. Strengths and Weaknesses

### a) Soundness

**Strengths.**
The technical framing is coherent: diffusion forcing is motivated as addressing rolling-window distribution shift and long-horizon instability, and the Bayesian decomposition of posterior score into prior + likelihood is clearly stated. The paper also makes a commendable effort to validate physical consistency via spectra (PSD), distributional comparisons, stability rollouts, and a conservation-guided variant.

**Weaknesses.**
The empirical story is sometimes uneven relative to the claim of replacing specialized baselines across a vast array of tasks. The paper frequently concedes that specialized models achieve lower RMSE (especially for downscaling) and leans on PSD/distributional alignment to argue superiority. That is a reasonable position, but it requires stronger justification that PSD fidelity correlates with end-user utility for the stated tasks, and more careful discussion of when pixel-wise errors are truly the wrong metric versus when they reflect materially wrong outcomes (e.g., precipitation localization).

### b) Presentation

**Strengths.** The narrative is strong and the train once/solve by inference paradigm is easy to follow. The paper clearly enumerates operators $A(\cdot)$ for tasks and provides experimental breakdowns and ablations (e.g., guidance vs free run, conservation).

**Weaknesses.** Some positioning claims could be more precise. For example, the work emphasizes conceptual novelty and versatility over competing with operational baselines due to compute limits, but the abstract and framing imply a broader replacement claim than the experiments fully substantiate. Also, the model design choice of not conditioning on noise level is a major deviation that deserves more prominent, intuitive explanation and ablation in the main narrative (currently easy to miss unless reading details).

### c) Significance
**Strengths.** A unified, zero-shot inference-only approach that supports constraint enforcement and counterfactual storylines is potentially impactful, particularly for research workflows where task-specific training is expensive and where physical constraints matter. The counterfactual section is particularly interesting as a demonstration of controllable generative modeling beyond forecasting accuracy.

**Weaknesses.** The current evaluation is limited by resolution choices $(1.5\degree)$ and by the acknowledged compute/inference cost of iterative sampling + MMPS. If the method is not computationally feasible for many real settings, significance shifts from “operational replacement to research tool/conceptual framework, which is still valuable but should be stated more cleanly.

### d) Originality
**Strengths.** The combination of diffusion forcing pretraining + posterior sampling/MMPS as a single foundation-model interface for heterogeneous tasks, without additional fine-tuning, is a meaningful and well-motivated synthesis. The operator-as-task abstraction is clear and extensible.

**Weaknesses.** Several individual components (diffusion forcing, posterior sampling, guidance, downscaling with diffusion) exist in prior work; originality rests on the unification and the breadth of demonstrated operators. To fully claim that unification as a contribution, the paper would benefit from clearer criteria about what qualifies as task replacement and where it does not.


## 3. Key Questions for Authors
1. Authors argue RMSE is inflated due to small spatial/temporal shifts while PSD fidelity remains high. Can you provide task-aligned metrics (e.g., feature-based precipitation object scores, spatial verification metrics, or event-based extremes metrics) to substantiate that your outputs are better for downscaling beyond spectra? How would that affect the headline conclusions?

2. Not conditioning on noise level is described as important. How sensitive are results to this design choice across tasks (forecasting vs inverse tasks)? A main-text ablation would materially affect confidence in the method.

3. What is the wall-clock inference cost per task (forecast rollout, downscaling, sparse reconstruction) and how does it scale with window length T, resolution, and number of MMPS CG steps? This directly impacts practical significance.

4. How robust is MMPS guidance when $A(\cdot)$ is imperfect or when observation noise is misspecified? Does the method fail gracefully or collapse into artifacts? This matters for real sensor settings.

5. The storyline experiment is compelling, but can you justify that the guidance does not introduce subtle dynamical inconsistencies beyond the reported SSIM/stationarity checks? What additional diagnostics would you recommend to domain scientists?

## 4. Limitations
The paper discusses key limitations notably inference speed and underestimation of extremes and outlines plausible future directions.

---

### Official Review · Reviewer_p1Gj · 2026-02-22
**Versatile atmospheric model**

**Rating:** 6
**Confidence:** 4

**Review:**

This authors introduce an atmospheric foundation model called WIND. The authors combine video diffusion forcing with MMPS to solve various tasks as inverse problems. The work also contains physics constraints to prevent unphysical drift While the model has impressive structural fidelity and stability, it suffers from high inference costs and does not match the raw pixel-wise accuracy of other baseline models.

In this work, I see limited algorithmic novelty. The core method relies heavily on existing techniques. The primary contribution is the application of established diffusion forcing and MMPS to ERA5 weather data, rather than a fundamental algorithmic breakthrough.

The model fails to outperform specialized, single-task baselines (like the task-specific UViT) on standard evaluation metrics such as RMSE for downscaling tasks. The zero-shot flexibility comes at a direct cost to raw predictive accuracy.

The model is trained and evaluated on a 1.5° spatial resolution. Operational models typically require 0.25° resolution. It remains unconvincing whether this method can scale effectively to higher resolutions.

I see the authors do not compare their forecasting capabilities against state-of-the-art models like GraphCast or GenCast. They rely on a custom proxy (AR-UViT), which makes the relative gains less impactful.

---

### Official Review · Reviewer_chHV · 2026-02-23
**Ambitious and original zero-shot atmospheric foundation modeling via diffusion-forcing + inverse problem guidance, with strong empirical breadth but some benchmarking and validation gaps**

**Rating:** 8
**Confidence:** 3

**Review:**

This paper introduces WIND, a unified atmospheric foundation model that is pre-trained once (via diffusion forcing on ERA5) and then used at inference time to solve a wide range of weather/climate tasks as inverse problems using moment matching posterior sampling (MMPS). The key claim is that the same pre-trained model can handle probabilistic forecasting, spatial downscaling, temporal downscaling, sparse reconstruction, conservation-law enforcement, and counterfactual storylines without task-specific fine-tuning.

Summary and overall assessment

I found the paper highly original and conceptually compelling, especially for the FM4Science workshop setting. The core idea—learning a general atmospheric prior and then solving downstream tasks through posterior sampling under task-specific operators
𝐴
A—is elegant and potentially impactful. The combination of:

diffusion forcing for flexible spatiotemporal conditioning,

zero-shot inverse-problem formulation,

and physically motivated guidance (e.g., dry air mass conservation)

is a strong contribution.

The empirical section is broad and ambitious, and the paper goes beyond standard pixel-wise metrics by including CRPS/SSR, power spectral density (PSD) analyses, long rollouts, and qualitative reconstructions. I also appreciate that the authors are explicit about the tradeoff: specialized baselines often have lower RMSE, while WIND offers versatility and better high-frequency structure preservation.

That said, several claims (especially around “replacing specialized baselines” and “physically consistent counterfactuals”) would benefit from stronger validation and more careful framing. The benchmark setup, comparison choices, and computational cost accounting leave open questions about practical competitiveness and fairness.


Strengths

1. The paper presents a genuinely unified perspective:
Y=A(X)+η,and sample from p(X∣Y)
using a single pre-trained model as the prior. This is a powerful abstraction that cleanly unifies multiple atmospheric tasks that are usually treated separately.

2. Clever use of diffusion forcing for flexible inference

Using independent per-frame noise levels during training is well-motivated and addresses a real weakness of standard video diffusion / full-sequence diffusion for rolling forecasts. The argument that this permits arbitrary combinations of clean/noisy context frames (and thus stable long rollouts / flexible conditioning) is compelling.

3. Impressive empirical breadth

The breadth of downstream tasks is a major strength:

probabilistic forecasting,
spatial downscaling,
temporal downscaling,
sparse reconstruction,
conservation-law enforcement,
counterfactual storyline generation.

It is rare to see a single framework evaluated across this many scientifically relevant settings.

4. Good use of physically meaningful diagnostics

The paper does not rely only on RMSE. The use of CRPS, SSR, PSD, distributional checks, long-term rollouts, and dry-air-mass conservation constraints gives a richer picture of model behavior and is appropriate for scientific ML.


Weaknesses:

1. “Zero-shot” is correct in one sense, but may be overstated in another

It is true that there is no task-specific fine-tuning, which is an important distinction. However, each task still requires:

a hand-designed operator 𝐴
an inverse-problem formulation,
likely task-specific guidance settings / MMPS hyperparameters,
and evaluation-specific setup choices.

So this is not “zero-effort adaptation.” I recommend the authors clarify this as zero-shot w.r.t. model parameter updates (no gradient-based adaptation), but not necessarily zero-shot w.r.t. task engineering.


2. Benchmark fairness / competitiveness is not fully convincing

The paper explicitly says it prioritizes conceptual novelty over SOTA operational competitiveness (reasonable), but several comparisons still feel limited:

Evaluation is at 1.5° ERA5 rather than higher-resolution operational settings.

Baselines are often internal/specialized variants rather than strongest current public baselines for each task.

For some tasks, the main comparison is to a deterministic model and one UViT variant; stronger conditional diffusion or task-optimized probabilistic baselines may exist.

This does not invalidate the contribution, but it weakens broad claims like “capable of replacing specialized baselines.”

3) Insufficient compute/runtime analysis

A major limitation acknowledged by the authors is inference speed, but the paper needs clearer quantitative reporting:

wall-clock inference time per sample / per forecast horizon,

cost of MMPS guidance vs unguided sampling,

number of DDIM steps and CG iterations (partially given),

hardware used for training and inference,

memory footprint / throughput.

Without this, it is hard to assess practical applicability for downstream scientific workflows.


4) Physical consistency claims are promising but still narrow

The paper demonstrates dry-air-mass (DAM) conservation and discusses physically plausible counterfactual behavior, which is exciting. However:

physical consistency is broader than DAM,

reanalysis is not strictly conservative,

and the paper itself notes missing flux variables prevent enforcing some constraints.

The claims should be framed more cautiously as selective physically guided consistency, not general physical correctness.

5) Counterfactual storyline section is exciting but needs stronger caveats

The Storm Bernd warm-world experiments are one of the most interesting parts, but also the most scientifically delicate. The paper would benefit from:

stronger methodological caveats distinguishing this from formal event attribution,

sensitivity analysis (operator choice, guidance strength, initialization perturbation magnitude),

more than one event case study,

uncertainty statements around causal interpretation.

As written, some readers may over-interpret the results.

6) Ablation coverage could be stronger

Several design choices are central to the paper, but I did not see enough ablation evidence in the provided text for:

no-noise-conditioning vs noise conditioning,

diffusion forcing vs standard video diffusion (beyond AR-UViT behavior),

MMPS vs simpler DPS-style likelihood guidance,

effect of sampling steps / guidance hyperparameters,

window length 𝑇

training resolution and variable subset.

These would strengthen the causal story behind the improvements.

7) Some reporting inconsistencies / clarity issues

I noticed a few places where reporting could be clearer (possibly resolved in appendices, but worth fixing):

Forecast evaluation mentions both 24 initial conditions and later references 100 initializations.

Some qualitative claims are strong relative to quantitative differences.

There are small wording/typo issues (e.g., grammar in contributions, “Bottowm,” etc.) that slightly affect polish.

Questions:


What hyperparameters are task-specific at inference time?

How sensitive is performance to MMPS settings?

Can you provide a standardized compute/runtime table?

---

### Meta-Review · Area_Chair_ZXeQ · 2026-02-27

**Recommendation:** Accept (Poster)
**Confidence:** 4

**Metareview:**

The average review score is above 6, which means reviewers recommended an acceptance.

---

### Decision · Program_Chairs · 2026-03-03

Accept (Poster)